# LobsDICE: Offline Learning from Observation via Stationary Distribution Correction Estimation

**Geon-Hyeong Kim**[1,3,*,†]**, Jongmin Lee**[4,*]**, Youngsoo Jang**[1,3,†]**,**
**Hongseok Yang**[1,2,5] **Kee-Eung Kim**[1,2]
[1] School of Computing, KAIST
[2] Kim Jaechul Graduate School of AI, KAIST
[3] LG AI Research
[4] University of California, Berkeley
[5] Discrete Mathematics Group, Institute for Basic Science (IBS)

## Abstract

We consider the problem of learning from observation (LfO), in which the agent aims to mimic the expert's behavior from the state-only demonstrations by experts. We additionally assume that the agent cannot interact with the environment but has access to the action-labeled transition data collected by some agents with unknown qualities. This offline setting for LfO is appealing in many real-world scenarios where the ground-truth expert actions are inaccessible and the arbitrary environment interactions are costly or risky. In this paper, we present LobsDICE, an offline LfO algorithm that learns to imitate the expert policy via optimization in the space of stationary distributions. Our algorithm solves a single convex minimization problem, which minimizes the divergence between the two state-transition distributions induced by the expert and the agent policy. Through an extensive set of offline LfO tasks, we show that LobsDICE outperforms strong baseline methods.

## 1 Introduction

The ability to learn from experience is one of the core aspects of an intelligent agent. Reinforcement learning (RL) [36] provides a framework to acquire such an intelligent behavior autonomously through interactions with the environment while receiving reward feedback. However, the practical applicability of RL to real-world domains has been limited for two reasons. First, designing a suitable reward function for complex tasks can be extremely difficult. The RL agent learns behaviors incentivized by the reward function rather than the ones intended, which can be nontrivial to specify in terms of reward. Second, the need for online interaction with the environment during the RL training loop has hindered its adoption in many real-world domains, where environment interactions are costly or risky.

Imitation learning (IL) [1, 33, 34] circumvents the difficulty of reward design in RL by leveraging demonstrations given by experts, where the goal is to mimic the expert's behavior. However, the standard IL requires the expert demonstrations to contain not only the state information (e.g. robot joint angles) but also the precise action (e.g. robot joint torques) executed by the expert at each time step. This demand for explicit action labels is in contrast to the way human imitates (e.g. learning by watching videos) and precludes leveraging a massive amount of data in which the action label is missing. Therefore, developing an imitation learning algorithm that can learn from observing the

---

[*]Equal contribution.
[†]Work done while the authors were students at KAIST.

36th Conference on Neural Information Processing Systems (NeurIPS 2022).

experts' state-only trajectories is a promising direction for creating a practical, autonomous intelligent agent. Learning from observation (LfO) [2, 14, 26, 39, 45] concerns this particular learning scenario and has been gaining interest in recent years.

In this paper, we are particularly interested in solving the LfO problem in an *offline* setting: Given the state-only demonstrations by experts and abundant state-action demonstrations by imperfect agents of arbitrary levels of optimality, we aim to find a policy that follows the sequence of expert's states without further interaction with the environment. This problem setting is appealing in many practical situations where environment interactions are costly or risky. Yet, it cannot be straightforwardly tackled by existing approaches. Most of the existing LfO methods are on-policy algorithms [25, 35, 39, 40], which is not directly applicable to the offline setting. One of the few exceptions is BCO [38], which performs behavior cloning on the inferred action via an inverse dynamics model. This method could be suboptimal unless the inverse dynamics disagreement is always zero, i.e. the underlying environment dynamics is deterministic and injective. Another one is OPOLO [45], an off-policy LfO algorithm, but it relies on nested min-max optimization as well as out-of-distribution (OOD) action values, which can be unstable especially in the offline setting. IQ-Learn [8] solves an Inverse RL and can be applied to the offline LfO by learning a state-only reward function. However, it suffers from numerical instability due to using OOD action values in the offline setting. Lastly, RCE [5], an example-based control algorithm, can in principle be applied to the offline LfO setting by providing the expert trajectories as successful example states; however, its empirical performance is known to be limited without online data collection.

We present an offline LfO algorithm that minimizes the divergence between state-transition distributions induced by the expert and the learned policy, without requiring an inverse dynamics model. Our algorithm, *offline Learning from OBServation via stationary DIstribution Correction Estimation* (LobsDICE), essentially optimizes in the space of state-action stationary distributions and state-transition stationary distributions, rather than in the space of policies. We show that our formulation can be reduced to a single convex minimization problem that can be solved efficiently in practice, unlike existing imitation learning (from observation) algorithms that rely on nested min-max optimization. In the experiments, we demonstrate that LobsDICE can successfully recover the state visitations by the expert, outperforming strong baseline methods.

## 2 Preliminaries

### 2.1 Markov decision process

We consider an environment modeled as a Markov Decision Process (MDP), defined by $M = \langle S, A, T, R, p_0, \gamma \rangle$ [36], where $S$ is the set of states, $A$ is the set of actions, $T : S \times A \to \Delta(S)$ is the transition probability, $R : S \times A \to \mathbb{R}$ is the reward function, $p_0 \in \Delta(S)$ is the distribution of the initial state, and $\gamma \in (0, 1)$ is the discount factor. The policy $\pi : S \to \Delta(A)$ is a mapping from states to distribution over actions. For the given policy $\pi$, its *state-action* stationary distribution $d^\pi(s, a)$ and *state-transition* stationary $\bar{d}^\pi(s, s')$ are defined as:

$$d^\pi(s, a) := (1 - \gamma) \sum_{t=0}^{\infty} \gamma^t \Pr(s_t = s, a_t = a), \quad \bar{d}^\pi(s, s') := (1 - \gamma) \sum_{t=0}^{\infty} \gamma^t \Pr(s_t = s, s_{t+1} = s'),$$

where $s_0 \sim p_0$, $a_t \sim \pi(\cdot|s_t)$, and $s_{t+1} \sim T(\cdot|s_t, a_t)$ for all timesteps $t \geq 0$. For brevity, the bar notation $\bar{(\cdot)}$ will be used to denote the distributions for $(s, s')$, e.g. $\bar{d}^\pi(s, s')$.

We assume offline LfO setting, where direct, online interactions with the environment are not allowed, and the policy should be optimized solely from a pre-collected dataset. We denote the dataset of state-only demonstrations collected by experts as $D^E = \{(s, s')_i\}_{i=1}^{N_E}$ and the dataset of state-action demonstrations by some imperfect agents as $D^I = \{(s, a, s')_i\}_{i=1}^{N_I}$. That is, we do not have information about the actions taken by the expert, but instead, we have additional action-labeled transition data collected by some other agents with unknown levels of optimality. We denote the corresponding distributions of the datasets $D^E$ and $D^I$ by $\bar{d}^E$ and $d^I$, respectively. For brevity, we will abuse notation $d^I$ to represent $(s, a) \sim d^I$, $(s, a, s') \sim d^I$, and $(s, s') \sim \bar{d}^I$ unless ambiguous.

## 2.2 Imitation learning and learning from observation

*Imitation learning* (IL) aims to mimic the expert policy from its state-action demonstrations. IL can be naturally formulated as a distribution matching problem that minimizes the divergence between *state-action* stationary distributions induced by the expert and the target policy [11, 13]. For example, one can consider minimizing KL-divergence [17]:

$$\min_{\pi} D_{\mathrm{KL}}(d^{\pi}(s,a)\|d^{E}(s,a)) = \mathbb{E}_{(s,a)\sim d^{\pi}}\left[\log \frac{d^{\pi}(s,a)}{d^{E}(s,a)}\right]. \tag{1}$$

However, the standard IL requires action labels in the expert demonstrations, which may be a too strong requirement for various practical situations. *Learning from observation* (LfO) relaxes the requirement on action labels, and aims to imitate the expert's behavior only from the state observations. Since the expert's action information is missing in the demonstrations, the distribution matching for state-action stationary distribution is no longer readily applicable. Therefore, LfO is reformulated as another distribution matching problem that minimizes the divergence between *state-transition* stationary distributions induced by the expert and the target policy [39, 40, 45][1]:

$$\min_{\pi} D_{\mathrm{KL}}(\bar{d}^{\pi}(s,s')\|\bar{d}^{E}(s,s')) = \mathbb{E}_{(s,s')\sim \bar{d}^{\pi}}\left[\log \frac{\bar{d}^{\pi}(s,s')}{\bar{d}^{E}(s,s')}\right]. \tag{2}$$

Still, optimizing Eq. (2) in a purely offline manner is challenging since naively estimating the expectation would require the knowledge of $T(\cdot|s,a)$ for the OOD action $a \sim \pi(s)$ by the target policy, which is inaccessible in the offline LfO setting; see Section 9.8 in [45] for more discussions. OPOLO [45], an off-policy LfO algorithm, mitigates this challenge by minimizing the following *upper bound* of the divergence $D_{\mathrm{KL}}(\bar{d}^{\pi}(s,s')\|\bar{d}^{E}(s,s'))$:

$$(2) \le \mathbb{E}_{\bar{d}^{\pi}(s,s')}\left[\log \frac{\bar{d}^{I}(s,s')}{\bar{d}^{E}(s,s')}\right] + D_{\mathrm{KL}}(d^{\pi}(s,a)\|d^{I}(s,a)) \tag{3}$$

and applying DualDICE trick [29] to the RHS of (3). The upper bound gap is given by the inverse dynamics disagreement between the target policy and the imperfect demonstrator:

$$(3) - (2) = D_{\mathrm{KL}}(d^{\pi}(a|s,s')\|d^{I}(a|s,s')), \tag{4}$$

which usually gets larger as the stochasticity of the environment transition increases.

## 3 LobsDICE

We present *offline Learning from OBServation via stationary DIstribution Correction Estimation* (LobsDICE), a principled offline LfO algorithm that further extends the recent progress made by the DIstribution Correction Estimation (DICE) methods for offline RL. LobsDICE essentially optimizes the stationary distributions of the target policy to match the expert's state visitations.

In the context of offline RL, the policy constraint principle (i.e. prevent deviating too much from the data support) is one of the common approaches to avoid severe performance degradation [7, 12, 18, 22, 30]. In the same manner, we use KL divergence minimization between $\bar{d}^{\pi}(s,s')$ and $\bar{d}^{E}(s,s')$ with additional KL regularization on the deviation from $d^{I}$:

$$\min_{\pi} D_{\mathrm{KL}}(\bar{d}^{\pi}(s,s')\|\bar{d}^{E}(s,s')) + \alpha D_{\mathrm{KL}}(d^{\pi}(s,a)\|d^{I}(s,a)). \tag{5}$$

Here the hyperparameter $\alpha > 0$ balances between encouraging state-transition matching and preventing distribution shift from the distribution of imperfect demonstrations. All the proofs can be found in Appendix F.

### 3.1 Lagrange dual formulation

The derivation of our algorithm starts by rewriting the (regularized) distribution matching problem (5) in terms of directly optimizing stationary distribution, rather than policy:

$$\max_{d,\bar{d}\ge 0} \; -D_{\mathrm{KL}}(\bar{d}(s,s')\|\bar{d}^{E}(s,s')) - \alpha D_{\mathrm{KL}}(d(s,a)\|d^{I}(s,a)) \tag{6}$$

$$\text{s.t.} \; \sum_{a'} d(s',a') = (1-\gamma)p_0(s') + \gamma \sum_{s,a} d(s,a)T(s'|s,a) \quad \forall s', \tag{7}$$

$$\sum_{a} d(s,a)T(s'|s,a) = \bar{d}(s,s') \quad \forall s,s', \tag{8}$$

---

[1]See Appendix A for a discussion of why $\bar{d}^{\pi}(s,s')$-matching is preferable to $\bar{d}^{\pi}(s)$-matching.

The Bellman flow constraint (7) ensures $d(s,a)$ to be a valid state-action stationary distribution of some policy, where $d(s,a)$ can be interpreted as a normalized occupancy measure of $(s,a)$. The marginalization constraint (8) enforces $\bar{d}(s,s')$ to be the state-transition stationary distribution that is directly induced by $d(s,a)$. In essence, the constrained optimization problem (6-8) seeks the stationary distributions of an optimal policy, which best matches the state-transition trajectories of the expert. Once we have computed the optimal solution $(d^*, \bar{d}^*)$, its corresponding optimal policy can also be obtained by normalizing $d^*$ for each state [32]: $\pi^*(a|s) = \frac{d^*(s,a)}{\sum_a d^*(s,a)}$.

Note that DemoDICE [15] considers a similar optimization problem to ours, but it deals with the offline IL (i.e. *state-action* stationary distribution matching), whereas we consider the offline LfO (e.g *state-transition* stationary distribution matching). Accordingly, the optimization variable $\bar{d}(s,s')$ and the marginalization constraint (8) are newly added in our formulation.

We then consider the Lagrangian for the constrained optimization (6-8):

$$\min_{\mu,\nu} \max_{d,\bar{d}\geq 0} - \mathbb{E}_{\bar{d}}\Big[\log \frac{\bar{d}(s,s')}{\bar{d}^E(s,s')}\Big] - \alpha\mathbb{E}_d\Big[\log \frac{d(s,a)}{d^I(s,a)}\Big] + \sum_{s,s'}\mu(s,s')\big(\bar{d}(s,s') - \sum_a d(s,a)T(s'|s,a)\big)$$
$$+ \sum_{s'}\nu(s')\big((1-\gamma)p_0(s') + \gamma\sum_{s,a}d(s,a)T(s'|s,a) - \sum_{a'}d(s',a')\big), \tag{9}$$

where $\nu(s) \in \mathbb{R}$ are the Lagrange multipliers for the Bellman flow constraints (7), and $\mu(s,s') \in \mathbb{R}$ are the the Lagrange multipliers for the marginalization constraint (8). Note that the Lagrangian (9) cannot be naively optimized in an offline manner since it requires evaluation of $T(s'|s,a)$ for $(s,a) \sim d$, which is not accessible in the offline LfO setting. Therefore, we rearrange the terms in (9) to eliminate the direct dependence on $d$ and $\bar{d}$, introducing new optimization variables $w$ and $\bar{w}$ that denote stationary distribution correction ratios for $(s,a)$ and $(s,s')$, respectively:

$$(9) = \min_{\mu,\nu} \max_{d,\bar{d}\geq 0}(1-\gamma)\mathbb{E}_{s_0\sim p_0}[\nu(s_0)] + \mathbb{E}_{(s,s')\sim\bar{d}}\Big[\mu(s,s') \overbrace{- \log \frac{\bar{d}(s,s')}{\bar{d}^I(s,s')}}^{=:\bar{w}(s,s')} \underbrace{+ \log \frac{\bar{d}^E(s,s')}{\bar{d}^I(s,s')}}_{=:r(s,s')}\Big]^{=-\log \frac{\bar{d}(s,s')}{\bar{d}^E(s,s')}}$$
$$+ \mathbb{E}_{(s,a)\sim d}\Big[\underbrace{\mathbb{E}_{s'}[-\mu(s,s') + \gamma\nu(s')] - \nu(s)}_{=:e_{\mu,\nu}(s,a)} - \alpha \underbrace{\log \frac{d(s,a)}{d^I(s,a)}}_{=:w(s,a)}\Big] \tag{10}$$

$$= \min_{\mu,\nu} \max_{w,\bar{w}\geq 0}(1-\gamma)\mathbb{E}_{s_0\sim p_0}[\nu(s_0)] + \mathbb{E}_{(s,s')\sim\bar{d}^I}\big[\bar{w}(s,s')\big(r(s,s') + \mu(s,s') - \log \bar{w}(s,s')\big)\big]$$
$$+ \mathbb{E}_{(s,a)\sim d^I}\big[w(s,a)\big(e_{\mu,\nu}(s,a) - \alpha\log w(s,a)\big)\big] =: \mathcal{L}(w,\bar{w},\mu,\nu). \tag{11}$$

Similar to the assumption of full coverage which is fairly standard across a broad set of recent offline RL approaches [19, 22, 27], we make a milder assumption that $\bar{d}^I(s,s') > 0$ whenever $\bar{d}^E(s,s') > 0$. This assumption is necessary to recover the expert's behavior successfully. We introduce the log ratio $r(s,s') = \log \frac{\bar{d}^E(s,s')}{\bar{d}^I(s,s')}$ in (10) to take an expectation under $\bar{d}^I$ (instead of $\bar{d}^E$), which is assumed to have a broader support than $\bar{d}^E$. This log ratio $r(s,s')$ can be easily estimated using a pretrained discriminator for two datasets $D^E$ and $D^I$, which will be explained in detail in the following section.

In summary, LobsDICE aims to solve the minimax optimization,

$$\min_{\mu,\nu} \max_{w,\bar{w}\geq 0} \mathcal{L}(w,\bar{w},\mu,\nu). \tag{12}$$

The optimal solution $(w^*, \bar{w}^*)$ of (12) represents stationary distribution corrections of an optimal policy $\pi^*$: $w^*(s,a) = \frac{d^{\pi^*}(s,a)}{d^I(s,a)}$ and $\bar{w}^*(s,s') = \frac{\bar{d}^{\pi^*}(s,s')}{\bar{d}^I(s,s')}$.

## 3.2 Log ratio estimation via a pretrained discriminator

To optimize (11), an estimate of the log ratio $r(s,s') = \log \frac{\bar{d}^E(s,s')}{\bar{d}^I(s,s')}$ is required. The log ratio estimation is straightforward for tabular MDPs since we can use empirical distributions from the datasets to estimate $\bar{d}^E(s,s')$ and $\bar{d}^I(s,s')$. For continuous MDPs, we train a discriminator $c : S \times S \to [0,1]$ by solving the following maximization problem [9, 45]:

$$c^* = \arg\max_{c:S\times S\to[0,1]} \mathbb{E}_{(s,s')\sim\bar{d}^E}[\log c(s,s')] + \mathbb{E}_{(s,s')\sim\bar{d}^I}[\log(1 - c(s,s'))]. \tag{13}$$

It is easy to show that the optimal discriminator satisfies $c^*(s, s') = \frac{\bar{d}^E(s,s')}{\bar{d}^E(s,s')+\bar{d}^I(s,s')}$. Thus, $r(s, s')$ can be derived from the optimal discriminator $c^*$ as,

$$r(s, s') = -\log\left(\frac{1}{c^*(s,s')} - 1\right). \tag{14}$$

### 3.3 Minimax to min: a closed-form solution

Exploiting the strict convexity of $x \log x$, we can derive a closed-form solution to the inner maximization for $(w, \bar{w})$ in (11).

**Proposition 3.1.** *For any $(\mu, \nu)$, the closed-form solution to the inner maximization of* (11)*, i.e.* $(w_{\mu,\nu}, \bar{w}_\mu) = \arg\max_{w,\bar{w}\geq 0} \mathcal{L}(w, \bar{w}, \mu, \nu)$*, is given by:*

$$w_{\mu,\nu}(s, a) = \exp\left(\tfrac{1}{\alpha}e_{\mu,\nu}(s, a) - 1\right) \text{ and } \bar{w}_\mu(s, s') = \exp(r(s, s') + \mu(s, s') - 1). \tag{15}$$

Based on the above result, we reduce the nested min-max optimization of (11) to a single minimization by plugging the closed-form solution $(w_{\mu,\nu}, \bar{w}_\mu)$ into $\mathcal{L}(w, \bar{w}, \mu, \nu)$:

$$\min_{\mu,\nu} \mathcal{L}(w_{\mu,\nu}, \bar{w}_\mu, \mu, \nu) = (1 - \gamma)\mathbb{E}_{s\sim p_0}[\nu(s)] + \mathbb{E}_{(s,s')\sim\bar{d}^I}\left[\exp\left(r(s, s') + \mu(s, s') - 1\right)\right]$$
$$+ \alpha\mathbb{E}_{(s,a)\sim d^I}\left[\exp\left(\tfrac{1}{\alpha}e_{\mu,\nu}(s, a) - 1\right)\right]. \tag{16}$$

We can even show that $\mathcal{L}(w_{\mu,\nu}, \bar{w}_\mu, \mu, \nu)$ is a convex function of $\mu$ and $\nu$.

**Proposition 3.2.** $\mathcal{L}(w_{\mu,\nu}, \bar{w}_\mu, \mu, \nu)$ *is convex with respect to $\mu$ and $\nu$.*

In short, by operating in the space of stationary distributions, offline LfO can, in principle, be resolved by solving a *convex minimization* problem. This is in contrast to the existing LfO algorithms, which typically involve either an adversarial training that optimizes the policy and the discriminator [14, 39, 40] or a nested min-max optimization for the policy and the critic [45].

### 3.4 Policy extraction

So far, we have derived an algorithm that essentially solves the state-transition distribution matching problem via convex minimization. However, we obtain $(\mu^*, \nu^*)$ as the solution of (16), instead of the optimal policy $\pi^*$. The remaining problem is to extract the optimal policy from $(\mu^*, \nu^*)$. The first step is to see that we can obtain the state-action stationary distribution correction $w_{\mu^*,\nu^*}(s, a) = \frac{d^{\pi^*}(s,a)}{d^I(s,a)}$ of the optimal policy from the closed-form solution (15). Among many possibilities to extract the policy from the state-action distribution correction, we adopt weighted behavior cloning (WBC):

$$\max_\pi \mathbb{E}_{(s,a)\sim d^{\pi^*}}[\log\pi(a|s)] = \mathbb{E}_{(s,a)\sim d^I}[w_{\mu^*,\nu^*}(s, a)\log\pi(a|s)] \tag{17}$$

which aims to maximize the predicted probability of actions chosen by optimal policy $\pi^*$. This is done by BC on the offline dataset $D^I$ where each sample $(s, a)$ is weighted by $w_{\mu^*,\nu^*}(s, a)$. For tabular MDPs, we can formally show that WBC extracts an optimal policy $\pi^*$ (Appendix C).

### 3.5 Practical algorithm with sample-based approximation

In practice, we estimate $\mathcal{L}(w, \bar{w}, \mu, \nu)$ in (11) using samples from distribution $d^I$. Let $\hat{\mathbf{E}}_{x\in D}[f(x)] := \frac{1}{|D|}\sum_{x\in D} f(x)$ be a Monte-Carlo estimate of $\mathbb{E}_{x\sim p}[f(x)]$ where $D = \{x_i\}_{i=1}^{|D|} \sim p$. We denote each sample $(s, a, s')$ in $D^I$ as $x$ for brevity.

$$\min_{\mu,\nu} \max_{w,\bar{w}\geq 0} \hat{\mathcal{L}}(w, \bar{w}, \mu, \nu) := (1 - \gamma)\hat{\mathbf{E}}_{s_0\in D_0}[\nu(s_0)] \tag{18}$$
$$+ \hat{\mathbf{E}}_{x\in D^I}\left[\bar{w}(s, s')\left(r(s, s') + \mu(s, s') - \log\bar{w}(s, s')\right) + w(s, a)\left(\hat{e}_{\mu,\nu}(s, a, s') - \alpha\log w(s, a)\right)\right]$$

where $\hat{e}_{\mu,\nu}(s, a, s') := -\mu(s, s') + \gamma\nu(s') - \nu(s)$ is a single-sample estimate of $e_{\mu,\nu}(s, a)$. Note that this sample-based objective function $\hat{\mathcal{L}}(w, \bar{w}, \mu, \nu)$ can be estimated only from samples in the offline dataset $D^I$ and is an unbiased estimator of $\mathcal{L}(w, \bar{w}, \mu, \nu)$ as long as every sample in $D^I$ was collected by interacting with the underlying MDP. To the best of our knowledge, this is the first

result to directly solve the state-transition distribution matching problem in a fully offline manner. In contrast, OPOLO [45] relies on the (potentially loose) *upper bound* of the objective in (3).

To reduce the minimax optimization to a single minimization, we apply the non-parametric closed-form solution for each sample $x = (s, a, s')$ in $D^I$:

$$\widehat{w}_{\mu,\nu}(x) = \exp\left(\frac{1}{\alpha}\hat{e}_{\mu,\nu}(s, a, s') - 1\right) \text{ and } \widehat{\bar{w}}_\mu(x) = \exp(r(s, s') + \mu(s, s') - 1). \qquad (19)$$

which is analogous to (15). Plugging this result into (18) yields a sample-based objective function for minimization[2]:

$$\min_{\mu,\nu} \widehat{\mathcal{L}}(\mu, \nu) = (1 - \gamma)\hat{\mathbf{E}}_{s_0 \in D_0}[\nu(s_0)] + \hat{\mathbf{E}}_{x \in D^I}\Big[\exp\big(r(s, s') + \mu(s, s') - 1\big) \qquad (20)$$
$$+ \alpha \exp\left(\frac{1}{\alpha}\hat{e}_{\mu,\nu}(s, a, s') - 1\right)\Big].$$

Still, the variable $\mu \in \mathbb{R}^{S \times S}$ is much higher dimensional than $\nu \in \mathbb{R}^S$. So, it works as the main bottleneck for the overall optimization. Fortunately, we can further simplify (20) by eliminating its dependence on $\mu$ via exploiting an additional closed-form solution.

**Proposition 3.3.** *For any $\nu$, the closed-form solution to the minimization* (20) *with respect to $\mu$, i.e.* $\mu_\nu = \arg\min_\mu \widehat{\mathcal{L}}(\mu, \nu)$, *is*

$$\mu_\nu(s, s') = \frac{1}{1+\alpha}\big(-\alpha r(s, s') + \gamma\nu(s') - \nu(s)\big). \qquad (21)$$

Using the above result in (20), we obtain the following minimization problem:

$$\min_{\widehat{\nu}} \widehat{\mathcal{L}}(\widehat{\nu}) = (1 - \gamma)\hat{\mathbf{E}}_{s_0 \in D_0}[\widehat{\nu}(s_0)] + (1 + \alpha)\hat{\mathbf{E}}_{x \in D^I}\Big[\exp\left(\frac{1}{1+\alpha}\widehat{A}_{\widehat{\nu}}(s, a, s') - 1\right)\Big], \qquad (22)$$

where $\widehat{A}_\nu(s, a, s') := r(s, s') + \gamma\nu(s') - \nu(s)$. The remaining issue is that optimizing (22) is not practical because $\exp(\cdot)$ often causes numerical instability and gradient explosion. To address this, we use a numerically-stable alternative of (22):

**Proposition 3.4.** *Let $\widetilde{\mathcal{L}}(\widetilde{\nu})$ be the function:*

$$\widetilde{\mathcal{L}}(\widetilde{\nu}) = (1 - \gamma)\hat{\mathbf{E}}_{s_0 \in D_0}[\widetilde{\nu}(s_0)] + (1 + \alpha)\log\hat{\mathbf{E}}_{x \in D^I}\Big[\exp\left(\frac{1}{1+\alpha}\widehat{A}_{\widetilde{\nu}}(s, a, s')\right)\Big]. \qquad (23)$$

*Then, $\min_{\widehat{\nu}} \widehat{\mathcal{L}}(\widehat{\nu}) = \min_{\widetilde{\nu}} \widetilde{\mathcal{L}}(\widetilde{\nu})$ holds. Also, $\widetilde{\mathcal{L}}(\widetilde{\nu})$ is convex with respect to $\widetilde{\nu}$.*

In order see why minimizing $\widetilde{\mathcal{L}}(\widetilde{\nu})$ no longer suffers from numerical instability, note that the gradient $\nabla_x \log \mathbb{E}_{x\sim p}[\exp(h(x))] = \mathbb{E}_{x\sim p}[\frac{\exp(h(x))}{\mathbb{E}_{\bar{x}\sim p}[\exp(h(\bar{x}))]}\nabla_x h(x)]$ normalizes $\exp(\cdot)$ by softmax and thus tames large numerical values. Finally, we can show the following connection between $\widehat{\nu}^*$ and $\widetilde{\nu}^*$:

**Proposition 3.5.** *Let $\widehat{V}$ and $\widetilde{V}$ be the sets $\arg\min_{\widehat{\nu}} \widehat{\mathcal{L}}(\widehat{\nu})$ and $\arg\min_{\widetilde{\nu}} \widetilde{\mathcal{L}}(\widetilde{\nu})$, respectively. Then, $\widetilde{V} = \{\widehat{\nu}^* + C \mid \widehat{\nu}^* \in \widehat{V}, C \in \mathbb{R}\}$ holds.*

Proposition 3.5 implies that an unnormalized stationary distribution corrections of an optimal policy would be obtained from $\widetilde{\nu}^* \in \widetilde{V}$.

$$\widehat{w}_{\mu_{\widehat{\nu}^*}, \widehat{\nu}^*}(x) = \exp\left(\frac{1}{\alpha}\big(-\mu_{\widehat{\nu}^*}(s, s') + \gamma\widehat{\nu}^*(s') - \widehat{\nu}^*(s)\big) - 1\right)$$
$$= \exp\left(\frac{1}{1+\alpha}\widehat{A}_{\widehat{\nu}^*}(s, a, s') - 1\right) \qquad \text{(by (21))}$$
$$\propto \exp\left(\frac{1}{1+\alpha}\widehat{A}_{\widetilde{\nu}^*}(s, a, s')\right) =: \widetilde{w}_{\widetilde{\nu}^*}(x) \qquad \text{(by Proposition 3.5)}$$

---

[2]In contrast to the results in the previous sections where (16) and (11) are identical, $\widehat{\mathcal{L}}(\mu, \nu)$ in (20) is an upper bound of $\max_{w,\bar{w}} \widehat{\mathcal{L}}(w, \bar{w}, \mu, \nu)$ in (18), due to applying *per-sample* closed-form solutions. While unbiasedness has been sacrificed at the cost of eliminating the nested optimization, it enables much stable optimization in practice. The upper bound gap vanishes when the transition dynamics are deterministic.

**Policy extraction**   We must take caution when using $\widetilde{w}_{\widetilde{\nu}^*}(x)$ since it is an unnormalized density ratio, i.e. $\mathbb{E}_{x \sim d^I}[\widetilde{w}_{\widetilde{\nu}^*}(x)] \neq 1$. Therefore, to extract a policy, we perform weighted BC using self-normalized importance sampling [31]:

$$\max_{\pi} \frac{\sum_{x \in D^I} \widetilde{w}_{\widetilde{\nu}^*}(x) \log \pi(a|s)}{\sum_{x \in D^I} \widetilde{w}_{\widetilde{\nu}^*}(x)}, \tag{24}$$

which completes the derivation of the practical version of LobsDICE. To sum up, LobsDICE solves $\widetilde{\nu}^* = \arg\min_{\widetilde{\nu}} \widetilde{\mathcal{L}}(\widetilde{\nu})$ of (23) via gradient descent, and extracts a policy via self-normalized weighted BC of (24). Pseudocode for LobsDICE can be found in Appendix G.2.

## 4   Related Work

**Learning from observation (LfO)**   Recent approaches for LfO are mostly *on-policy* [25, 26, 35, 39, 40] algorithms and are not directly applicable to the offline LfO setting considered in this work. MobILE [14] is a model-based LfO algorithm, but it encourages uncertainty for online exploration, which is not suitable for the offline setting. BCO [38] uses an inverse dynamics model (IDM) to infer the missing expert actions and performs BC on the generated expert's state-action dataset. In addition to common issues by vanilla BC, BCO is not guaranteed to recover the expert's behavior in general. OPOLO [45] is a principled off-policy LfO algorithm, but it solves a nested optimization and requires evaluation on OOD action values during training, which suffers from numerical instability in the offline setting. IQ-Learn [8] solves online and offline IL problems while avoiding adversarial training by learning a single Q-function. However, it also suffers from numerical instability in the offline setting by overestimating $Q$ due to using OOD action values. RCE [5] aims to solve example-based control tasks in which a collection of example success states is assumed to be provided instead of entire expert trajectories. While RCE can be applied to LfO in principle, it tends to stay in a few expert states that are easy to reach in the offline setting, as discussed in Section C.2 of [5].

**Stationary DIstribution Correction Estimation (DICE)**   DICE-family algorithms perform stationary distribution estimation, and many of them have been proposed for off-policy evaluation [4, 29, 42–44]. Other lines of works consider reinforcement learning [22, 23, 30], offline policy selection [41]. ValueDICE [17] and OPOLO [45] derive off-policy IL and LfO objectives using DICE. However, they suffer from numerical instability in the offline setting due to the nested min-max optimization and OOD action evaluation. DemoDICE [15] is an offline IL algorithm that directly optimizes stationary distribution as ours and reduces to solving a convex minimization. Yet, it requires expert action labels and is not directly applicable to LfO. Concurrent to our work, SMODICE [28] is an offline LfO method, aiming to match stationary *state* distributions by minimizing an objective similar to OPOLO [45] in (3). Thus it also relies on potentially loose upper bound of the divergence. In addition, matching the state distributions may not be sufficient to recover the expert's behavior, on which we provide detailed discussion in Appendix A and Remark B.1.

## 5   Experiments

In this section, we evaluate LobsDICE on both tabular and continuous MDPs. We use four baseline methods in tabular MDPs: BC on imperfect demonstrations, BCO [38], and OPOLO [45]. Additionally, we designed a strong baseline DemoDICEfO, which extends the state-of-the-art offline IL algorithm, DemoDICE [15]. DemoDICEfO trains an inverse dynamics model, uses it to fill the missing actions in the expert demonstrations, and then runs DemoDICE using the approximate expert demonstrations and the imperfect demonstrations. For continuous control tasks, we use two additional baselines: IQ-Learn [8] and RCE [5].

### 5.1   Random MDPs

We first evaluate LobsDICE and baseline algorithms on randomly generated finite MDPs using a varying number of expert/imperfect trajectories and different degrees of stochasticity in the environment . We follow the experimental protocol in previous offline RL works [20–22] but with additional control on the stochasticity of transition probabilities. We conduct repeated experiments for 1K runs. For each run, (1) a random MDP is generated, (2) expert trajectories and imperfect trajectories are collected, and (3) each offline LfO algorithm is tested on the collected offline dataset. We evaluate

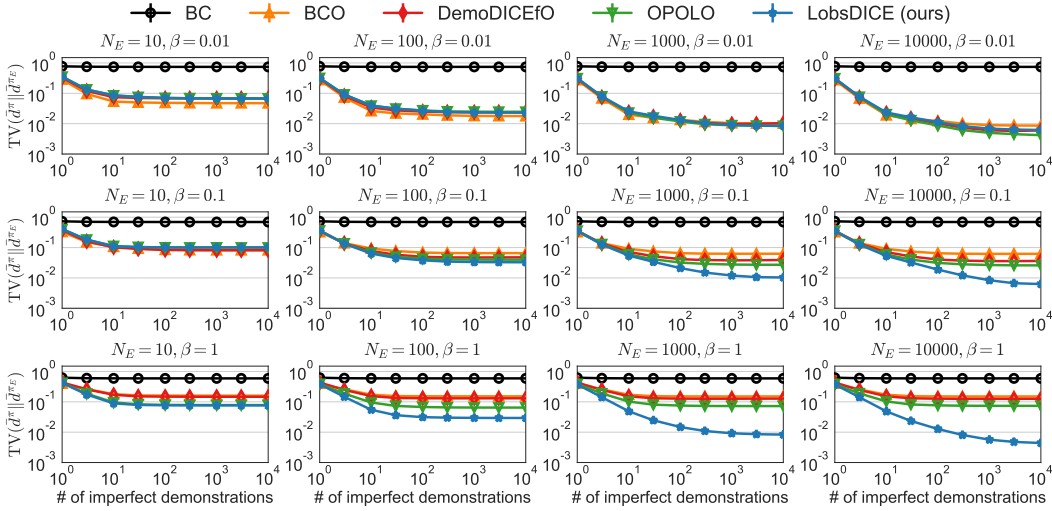

Figure 1: Performance of tabular LobsDICE and baselines in randomly generated MDPs. The first row indicates near-deterministic dynamics and the last row indicates highly stochastic dynamics. As the level of stochasticity increases, baselines fall into suboptimal, even the number of state-only expert demonstrations and imperfect demonstrations increases, while LobsDICE goes to optimal. For each algorithm, we measure the performance using total variation between state-transition stationary distributions of expert and learned policy. We plot the mean and standard error of total variations $\mathrm{TV}(\bar{d}^{\pi}(s,s'), \bar{d}^{\pi_E}(s,s'))$ over 1000 random seeds.

the performance of each algorithm by measuring the total variation distance between state-transition stationary distributions by the expert policy and the learned policy, i.e. $D_{\mathrm{TV}}(\bar{d}^{\pi}(s,s')\|\bar{d}^{\pi_E}(s,s'))$. For the tabular MDP experiments, we adopt tabular methods but not function approximation. Our tabular LobsDICE optimizes (16) on the empirical MDP model constructed from the action-labeled dataset $D^I$ while extracting the policy through (17). The pseudocode for tabular LobsDICE can be found in Appendix G.1. The detailed experimental setup such as random MDP generation and offline dataset generation can be found in Appendix I.1.

**Results** Figure 1 presents the results in random MDP experiments, where $\beta$ is the hyperparameter that controls the stochasticity of the underlying MDP. The first row corresponds to the case when $\beta = 0.01$ (nearly deterministic MDP). In this situation, the inverse dynamics disagreement will be close to zero, i.e. $D_{\mathrm{KL}}(d^{\pi_1}(a|s,s')\|d^{\pi_2}(a|s,s')) \approx 0$ for any two policies $\pi_1$ and $\pi_2$. Thus, the algorithms whose performance directly relies on the IDM's accuracy (i.e. BCO and DemoDICEfO) even perform very well since it is very easy to learn a perfect inverse dynamics model in this scenario. OPOLO's upper bound gap (4) will also be close to zero, thus OPOLO directly minimizes the divergence of state-transition distributions. As a result, there is no performance gap among different algorithms, except for BC whose performance is determined by the quality of imperfect demonstrations. Also, the performance of all algorithms (except for BC) improves as more data is given, which is natural.

The second row and the third row in Figure 1 presents the result when $\beta = 0.1$ (weakly stochastic MDP) and $\beta = 1.0$ (highly stochastic MDP) respectively. In the stochastic MDPs, the IDM trained by the *imperfect* demonstrations faces a challenge in predicting the expert's actions accurately (more challenging as $\beta$ gets larger). As a consequence, BCO gets suboptimal and its suboptimality cannot be improved even if more data is given. DemoDICEfO performs better than BCO since it additionally considers the distributional shift by considering state distribution matching. However, it is still suboptimal due to its nature that directly depends on the quality of inferred action by the learned IDM. OPOLO does not rely on the learned IDM and outperforms both BCO and DemoDICEfO. Still, OPOLO can be inherently suboptimal due to its nature of optimizing the *upper bound* unless the underlying transition dynamics are deterministic and injective. This upper bound gap (4) is not controllable by the algorithm and implies that OPOLO can be suboptimal even given an infinite

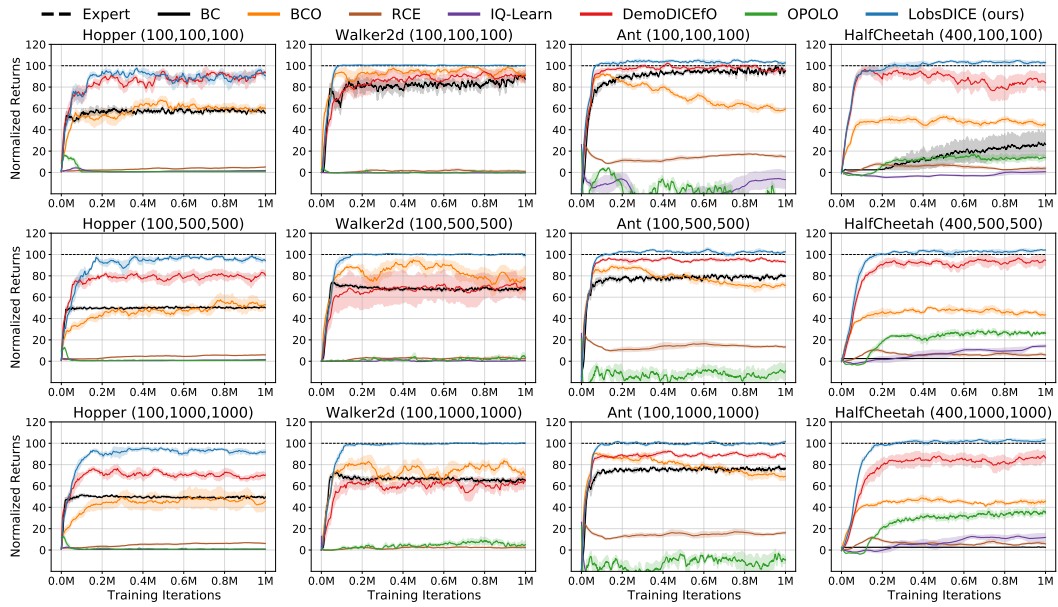

Figure 2: Performance of LobsDICE and baseline algorithms on various MuJoCo control tasks. We build state-only expert demonstrations using 5 trajectories from `expert-v2`. For each task $(X, Y, Z)$ we construct imperfect demonstrations using $X$, $Y$, and $Z$ trajectories from `expert-v2`, `medium-v2`, and `random-v2`, respectively. We plot the mean and the standard errors (shaded area) of the normalized scores over five random seeds.

amount of data with sufficient dataset coverage, which can be seen in the rightmost figures. Finally, our tabular LobsDICE using (16) essentially solves the exact state-transition distribution matching problem (as $\alpha \to 0$). LobsDICE is the only offline LfO algorithm that can asymptotically recovers the expert's state demonstrations even though the underlying MDP is stochastic.

## 5.2 Continuous control tasks (Gym-MuJoCo)

We present the empirical performance of LobsDICE and baselines on MuJoCo [37] continuous control tasks using the OpenAI Gym [3] framework. We utilize the D4RL dataset [6] for offline LfO tasks in four MuJoCo environments: Hopper, Walker2d, HalfCheetah, and Ant. Implementation details for LobsDICE and baseline algorithms such as hyperparameters and evaluation metric are provided in Appendix I.2.

**Task setup** For each MuJoCo environments, we employ `expert-v2`, `medium-v2`, and `random-v2` from D4RL datasets [6]. Across all environments, we consider three tasks, each of which uses different imperfect demonstrations while sharing the same expert observations. First, we construct the state-only expert demonstration $D^E = \{(s, s')_i\}_{i=1}^{N_E}$ using the first 5 trajectories in `expert-v2`. Then, we use trajectories in `expert-v2`, `medium-v2`, and `random-v2` to construct imperfect demonstrations with different ratios. We denote the composition of imperfect demonstrations as $(X, Y, Z)$ in the title of each subplot in Figure 2, which means that the imperfect dataset consists of $X$ trajectories from `expert-v2`, $Y$ trajectories from `medium-v2`, and $Z$ trajectories from `random-v2`.

**Results** Figure 2 summarizes that the empirical results of LobsDICE and baselines on continuous control tasks. We first remark that LobsDICE (blue) significantly outperforms OPOLO (green) in all tasks across all domains, although both LobsDICE and OPOLO are DICE-based algorithms. The failure of OPOLO comes from its numerical instability due to its dependence on nested optimization and using OOD action values during training. In contrast, LobsDICE solves a single minimization (23) while it does not involve any evaluation on OOD actions, thus it is optimized stably. IQ-Learn (purple) also suffers from numerical instability due to the usage of OOD action values during training, showing poor performance similar to OPOLO. RCE (brown) tends to stay in a few states that are easy to reach

in the offline setting, rather than following the entire expert trajectories. Naive BC on imperfect demonstrations (black) is inherently suboptimal since it does not consider distribution-matching with the expert's observation at all. While BCO (orange) exploits the expert's demonstrations with the inferred actions by the IDM, its policy learning is done only on the very scarce expert dataset (i.e. 5 trajectories), which makes the algorithm perform not well. DemoDICEfO (red) exploits both expert demonstrations (where the missing actions are filled with the IDM) and the abundant imperfect demonstrations, but its performance is affected by the quality of the learned IDM. We empirically observe that the IDM error (on the true expert data) increases as the proportion of the non-expert data (i.e. `medium-v2` and `random-v2`) increases, resulting in performance degradation of both BCO and DemoDICEfO. Finally, LobsDICE is the only algorithm that was able to fully recover the expert's performance regardless of the increase of non-expert data in the imperfect demonstrations, significantly outperforming baseline algorithms. This result highlights the effectiveness of our method that solves a state-transition stationary matching problem in a principled manner. We provide additional experiments in Appendix H.

## 6 Conclusion

We presented LobsDICE, an algorithm for offline learning from observations (LfO), which successfully achieves state-of-the-art performance on various tabular and continuous tasks. We formulated the offline LfO as a state-transition stationary distribution matching problem, where the stationary distribution is optimized via convex minimization. Experimental results demonstrated that LobsDICE achieves promising performance in both tabular and continuous offline LfO tasks.

## Acknowledgments and Disclosure of Funding

This work was supported by National Research Foundation (NRF) of Korea (NRF-2019R1A2C1087634), Field-oriented Technology Development Project for Customs Administration through National Research Foundation (NRF) of Korea funded by the Ministry of Science & ICT and Korea Customs Service (NRF-2021M3I1A1097938), Institute of Information & communications Technology Planning & Evaluation (IITP) grant funded by the Korea government (MSIT) (No.2020-0-00940, Foundations of Safe Reinforcement Learning and Its Applications to Natural Language Processing; No.2022-0-00311, Development of Goal-Oriented Reinforcement Learning Techniques for Contact-Rich Robotic Manipulation of Everyday Objects; No.2019-0-00075, Artificial Intelligence Graduate School Program (KAIST); No.2021-0-02068, Artificial Intelligence Innovation Hub), and Electronics and Telecommunications Research Institute (ETRI) grant funded by the Korean government (22ZS1100, Core Technology Research for Self-Improving Integrated Artificial Intelligence System). Hongseok Yang was supported by the Engineering Research Center Program through the National Research Foundation of Korea (NRF) funded by the Korean Government MSIT (NRF-2018R1A5A1059921) and also by the Institute for Basic Science (IBS-R029-C1). Kee-Eung Kim was supported by KAIST-NAVER Hypercreative AI Center.

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
