## A    Why State-transition Occupancy Matching Instead of State Occupancy Matching?

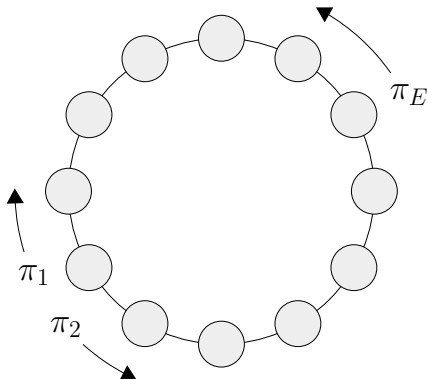

Figure 3: An example MDP with $|S| = 12$, $|A| = 2$ and $p_0(s) = \text{Unif}(s)$. In this example MDP, $\{\pi_1, \pi_2\} \subseteq \arg\min_\pi D_{\text{KL}}(d^\pi(s)||d^{\pi_E}(s))$, while $\{\pi_2\} = \arg\min_\pi D_{\text{KL}}(d^\pi(s,s')||d^{\pi_E}(s,s'))$.

For learning from observation, we adopt the objective function for $d(s, s')$-matching (i.e., $\min_\pi D_{\text{KL}}(d^\pi(s,s')||d^E(s,s')))$ instead of $d(s)$-matching (i.e. $\min_\pi D_{\text{KL}}(d^\pi(s)||d^E(s)))$ since $d(s)$-matching may ignore the *directionality* of the expert trajectories. To see this, consider an MDP with $|S|$ states and 2 actions, where states are denoted as circles in Figure 3. Each action moves the agent from the current state to the neighboring state deterministically either clockwise or counterclockwise. Initial state distribution $p_0$ is defined as uniform distribution over the entire states. Finally, let an expert $\pi_E$ be a policy that moves in a counterclockwise direction, and we want to mimic this expert's behavior. In this example, the desired imitation policy should be $\pi_2$ in Figure 3. However, if we perform $d(s)$-matching, even $\pi_1$ can be obtained as a resulting policy, which is the complete opposite of what we wanted to obtain. This is due to the fact that $\pi_1$ and $\pi_2$ share the same state stationary distribution of a uniform distribution: $d^\pi(s) = \frac{1}{|S|} \forall s$. Thus, both $\pi_1$ and $\pi_2$ must be an optimal solution of $\min_\pi D_{\text{KL}}(d^\pi(s)||d^E(s))$. In contrast, the $d(s, s')$-matching has a unique solution of $\pi_2$, exploiting the directionality information (i.e. moving counterclockwise) of the expert's trajectories.

## B    LobsDICE for State Occupancy Matching

While we have provided a counter-example where $d(s)$-matching can fail to recover the expert's behavior in Appendix A, we still can derive an offline LfO algorithm for $d(s)$-matching. For the given policy $\pi$, its state stationary distribution is defined as:

$$\bar{d}^\pi(s) := (1 - \gamma) \sum_{t=0}^\infty \gamma^t \Pr(s_t = s).$$

Then, similar to the state-transition occupancy matching objective (5), we formulate a state occupancy matching problem as follows:

$$\min_\pi D_{\text{KL}}(\bar{d}^\pi(s)||d^E(s)) + \alpha D_{\text{KL}}(d^\pi(s,a)||d^I(s,a)). \tag{25}$$

where the hyperparameter $\alpha > 0$ balances between encouraging state matching and preventing distribution shift from the distribution of imperfect demonstrations. This objective can be reformulated in terms of optimizing stationary distribution:

$$\max_{d, \bar{d} \geq 0} \ - D_{\text{KL}}(\bar{d}(s)||d^E(s)) - \alpha D_{\text{KL}}(d(s,a)||d^I(s,a)) \tag{26}$$

$$\text{s.t.} \ \sum_{a'} d(s', a') = (1 - \gamma)p_0(s') + \gamma \sum_{s,a} d(s,a)T(s'|s,a) \quad \forall s', \tag{27}$$

$$\sum_a d(s,a) = \bar{d}(s) \quad \forall s, \tag{28}$$

Then, we consider the Lagrangian for the constrained optimization (26-28):

$$\min_{\mu,\nu} \max_{d,\bar{d}\geq 0} -\mathbb{E}_{\bar{d}}\Big[\log \frac{\bar{d}(s)}{d^E(s)}\Big] - \mathbb{E}_d\Big[\log \frac{d(s,a)}{d^I(s,a)}\Big] + \sum_s \mu(s)\big(\bar{d}(s) - \sum_a d(s,a)\big) \tag{29}$$

$$+ \sum_{s'} \nu(s')\big((1-\gamma)p_0(s') + \gamma \sum_{s,a} d(s,a)T(s'|s,a) - \sum_{a'} d(s',a')\big)$$

where $\nu(s') \in \mathbb{R}$ are the Lagrange multipliers for the Bellman flow constraints (27), and $\mu(s) \in \mathbb{R}$ are the Lagrange multipliers for the marginalization constraints (28). To make the optimization tractable in the offline setting, we rearrange the terms in (29), introducing new optimization variables $w$ and $\bar{w}$ that denote stationary distribution correction ratios for $(s,a)$ and $s$ respectively.

$$(29) = \min_{\mu,\nu} \max_{d,\bar{d}\geq 0} (1-\gamma)\mathbb{E}_{s_0\sim p_0}[\nu(s_0)] + \mathbb{E}_{s\sim \bar{d}}\Big[\mu(s) - \underbrace{\log \frac{\bar{d}(s)}{d^I(s)}}_{=:\bar{w}(s)} + \underbrace{\log \frac{d^E(s)}{d^I(s)}}_{=:r(s)}\Big]$$

$$+ \mathbb{E}_{(s,a)\sim d}\Big[\underbrace{-\mu(s) + \mathbb{E}_{s'}[\gamma\nu(s')] - \nu(s)}_{=:e_{\mu,\nu}(s,a)} -\alpha \log \underbrace{\frac{d(s,a)}{d^I(s,a)}}_{=:w(s,a)}\Big]$$

$$= \min_{\mu,\nu} \max_{w,\bar{w}\geq 0} (1-\gamma)\mathbb{E}_{s_0\sim p_0}[\nu(s_0)] + \mathbb{E}_{s\sim d^I}\big[\bar{w}(s)\big(r(s) + \mu(s) - \log \bar{w}(s)\big)\big]$$

$$+ \mathbb{E}_{(s,a)\sim d^I}\big[w(s,a)\big(e_{\mu,\nu}(s,a) - \alpha \log w(s,a)\big)\big] =: \mathcal{L}(w,\bar{w},\mu,\nu). \tag{30}$$

We introduce the log ratio $r(s) = \log \frac{d^E(s)}{d^I(s)}$ to make expectation for $d^I$ instead of $d^E$. The log ratio $r(s)$ can be estimated by using a pretrained discriminator $c: S \to [0,1]$, where $c$ is trained by:

$$c^* = \arg\max_{c:S\to[0,1]} \mathbb{E}_{s\sim d^E}[\log c(s)] + \mathbb{E}_{s\sim d^I}[\log c(s)].$$

Then, it is proven that the optimal discriminator $c^*$ satisfies $c^*(s) = \frac{d^E(s)}{d^E(s)+d^I(s)}$. Therefore, $r(s)$ can be estimated by:

$$r(s) = -\log\left(\frac{1}{c^*(s)} - 1\right).$$

Similar to Proposition 3.1, we can easily derive the closed-form solution to the inner maximization in (30):

$$w_{\mu,\nu}(s,a) = \exp\left(\frac{1}{\alpha}e_{\mu,\nu}(s,a) - 1\right),$$

$$\bar{w}_\mu(s) = \exp(r(s) + \mu(s) - 1).$$

Finally, by plugging these closed-form solutions into (30), the nested min-max optimization of (30) is reduced to a single minimization:

$$\min_{\mu,\nu}\mathcal{L}(w_{\mu,\nu},\bar{w}_\mu,\mu,\nu) = (1-\gamma)\mathbb{E}_{s_0\sim p_0}[\nu(s_0)] + \mathbb{E}_{s\sim d^I}\big[\exp\big(r(s) + \mu(s) - 1\big)\big] \tag{31}$$

$$+ \alpha\mathbb{E}_{(s,a)\sim d^I}\Big[\exp\left(\frac{1}{\alpha}e_{\mu,\nu}(s,a) - 1\right)\Big],$$

which is a convex function of $\mu$ and $\nu$. Finally, once we obtain the optimal solution $(\mu^*, \nu^*)$ of (31), we have $w_{\mu^*,\nu^*}(s,a) = \frac{d^{\pi^*}(s,a)}{d^I(s,a)}$. Then, we can extract an optimal policy via weighted BC:

$$\max_\pi \mathbb{E}_{(s,a)\sim d^{\pi^*}}[\log \pi(a|s)] = \mathbb{E}_{(s,a)\sim d^I}[w_{\mu^*,\nu^*}(s,a) \log \pi(a|s)] \tag{32}$$

**Remark B.1.** To the best of our knowledge, (31) is the first result to directly solve the state distribution matching problem in an offline setting. Although SMODICE [28], a concurrent work with ours, also aims to solve offline LfO in terms of optimizing stationary distribution, it essentially minimizes the (potentially loose) *upper bound*, i.e.,

$$\min_\pi \mathbb{E}_{s\sim d^\pi}\left[\log \frac{d^I(s)}{d^E(s)}\right] + D_{\mathrm{KL}}(d^\pi(s,a)||d^I(s,a)) \tag{33}$$

$$\geq \mathbb{E}_{s\sim d^\pi}\left[\log \frac{d^I(s)}{d^E(s)}\right] + D_{\mathrm{KL}}(d^\pi(s)||d^I(s)) \tag{34}$$

$$= D_{\mathrm{KL}}(d^\pi(s)||d^E(s)) \tag{35}$$

The upper bound gap (33) − (35) is given by $D_{\mathrm{KL}}(\pi(a|s)\|\pi^I(a|s))$. Unlike OPOLO of (4), this upper bound gap does not vanish even when the transition is deterministic. Consequently, SMODICE may not be able to precisely recover the expert's state visitations even given an infinite amount of data due to the upper bound gap unless $d^I$ is collected purely by expert. In contrast, we are directly minimizing the divergence: solving (31) is exactly equivalent to solving (25). As can be seen in Figure 4, SMODICE optimizing (33) fails to recover the expert's behavior due to the upper bound gap that is irreducible even for deterministic MDPs.

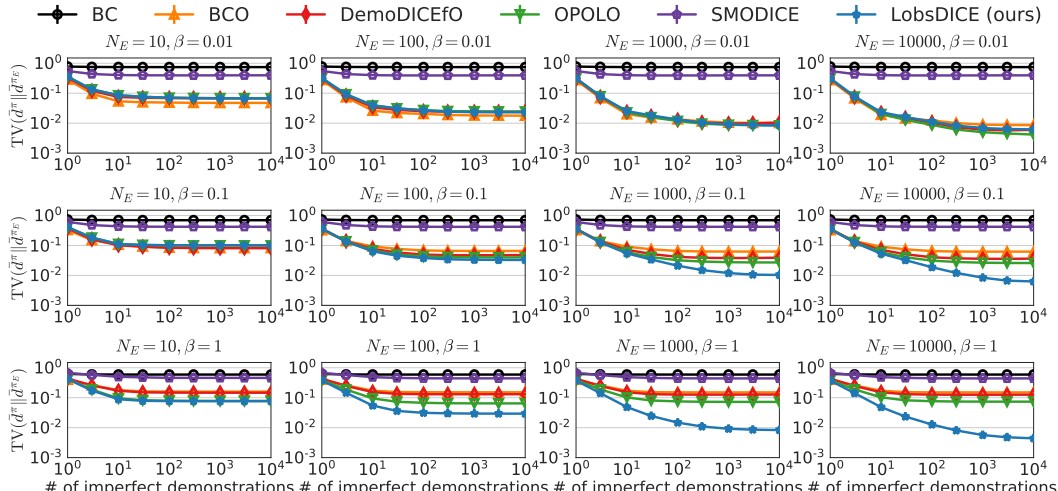

Figure 4: Performance of tabular LobsDICE and baselines in randomly generated MDPs. The first row indicates near-deterministic dynamics and the last row indicates highly stochastic dynamics. As the level of stochasticity increases, baselines fall into suboptimal, even the number of state-only expert demonstrations and imperfect demonstrations increases, while LobsDICE goes to optimal. For each algorithm, we measure the performance using total variation between state-transition stationary distributions of expert and learned policy. We plot the mean and standard error of total variations $\mathrm{TV}(\bar{d}^\pi(s, s'), \bar{d}^{\pi_E}(s, s'))$ over 1000 random seeds.

## C  Validity of Weighted Behavior Cloning for Policy Extraction

For tabular MDPs, we can show that the weighted behavior cloning of (17)

$$\max_\pi \mathbb{E}_{(s,a)\sim d^I}\left[w_{\mu^*,\nu^*}(s, a)\log \pi(a|s)\right] \tag{17}$$

exactly yields an optimal policy $\pi^*$ of the original optimization problem (5). First, the optimization problem (5) and the constrained optimization problem (6-8) are equivalent with the following relationship between $\pi^*$ and $d^*$:

$$\pi^*(a|s) = \frac{d^*(s, a)}{\sum_{a'} d^*(s, a')} \tag{36}$$

where $d^*$ is the optimal solution of both (6-8) and (10). Then, $w_{\mu^*,\nu^*}(s, a)$ is computed by (15) with $(\mu^*, \nu^*)$, the optimal solution of both (16) and (11). Consequently, $w_{\mu^*,\nu^*}(s, a) = \frac{d^*(s,a)}{d^I(s,a)}$ holds, and thus the weighted behavior cloning of (17) essentially performs the following:

$$\max_x \sum_{s,a} d^*(s, a)\log x(a|s) \tag{37}$$

$$\text{s.t.} \ \sum_a x(a|s) = 1 \quad \forall s \tag{38}$$

Now, consider the Lagrangian for the constrained optimization (37-38):

$$L := \sum_{s,a} d^*(s, a)\log x(a|s) - \sum_s \lambda(s)\Big(\sum_a x(a|s) - 1\Big) \tag{39}$$

where $\lambda(s)$ is the Lagrange multiplier for the constraint (38). Then,

$$\frac{\partial L}{\partial x(a|s)} = \frac{d^*(s,a)}{x(a|s)} - \lambda(s) = 0 \quad \Rightarrow x^*(a|s) = \frac{d^*(s,a)}{\lambda(s)} \ \forall s, a \tag{40}$$

Finally, considering the constraint $\sum_a x^*(a|s) = 1 \ \forall s$, we obtain:

$$x^*(a|s) = \frac{d^*(s,a)}{\sum_{a'} d^*(s,a')} \tag{41}$$

which is identical to (36). In summary, the weighted behavior cloning performs the process of extracting the optimal policy $\pi^*$ from $w_{\mu^*,\nu^*}$.

## D  Challenges of Extending DemoDICE to Offline Learning from Observation

DemoDICE [15] is an algorithm that optimizes the state-action stationary distribution $d(s,a)$, where its naive application to $(s,s')$-distribution matching problem yields:

$$\min_{d \geq 0} \mathbb{E}_{\substack{(s,a)\sim d \\ s'\sim T(s,a)}} \overbrace{\left[ \log \frac{\sum_a d(s,a)T(s'|s,a)}{d^E(s,s')} \right]}^{=D_{\mathrm{KL}}(\bar{d}(s,s')||\bar{d}^E(s,s'))} + \alpha \overbrace{\mathbb{E}_{(s,a)\sim d}\left[ \log \frac{d(s,a)}{d^I(s,a)} \right]}^{=D_{\mathrm{KL}}(d(s,a)||d^I(s,a))} \tag{42}$$

$$\text{s.t.} \sum_{a'} d(s',a') = (1-\gamma)p_0(s') + \gamma \sum_{s,a} d(s,a)T(s'|s,a) \quad \forall s', \tag{43}$$

However, estimating $\mathbb{E}_{\substack{(s,a)\sim d \\ s'\sim T(s,a)}}\left[ \log \frac{\sum_a d(s,a)T(s'|s,a)}{d^E(s,s')} \right]$ is intractable in the offline setting due to its inclusion of marginalization over $a$ inside the $\log(\cdot)$. This challenge can be addressed by using the *upper bound* proposed by OPOLO [45]:

$$D_{\mathrm{KL}}(\bar{d}^\pi(s,s')||d^E(s,s') \leq \mathbb{E}_{(s,s')\sim\bar{d}^\pi(s,s')}\left[ \log \frac{\bar{d}^I(s,s')}{\bar{d}^E(s,s')} \right] + D_{\mathrm{KL}}(d^\pi(s,a)||d^I(s,a)), \tag{3}$$

Replacing $D_{\mathrm{KL}}(\bar{d}^\pi(s,s')||d^E(s,s'))$ with the (potentially loose) *upper bound* in the objective function yields the following tractable (but biased) optimization problem:

$$\min_{d \geq 0} \mathbb{E}_{\substack{(s,a)\sim d \\ s'\sim T(s,a)}}\left[ \log \frac{\bar{d}^I(s,s')}{\bar{d}^E(s,s')} \right] + (1+\alpha)\mathbb{E}_{(s,a)\sim d}\left[ \log \frac{d(s,a)}{d^I(s,a)} \right] \tag{44}$$

$$\text{s.t.} \sum_{a'} d(s',a') = (1-\gamma)p_0(s') + \gamma \sum_{s,a} d(s,a)T(s'|s,a) \quad \forall s', \tag{45}$$

In contrast, we address the intractability challenge by introducing an additional optimization variable $\bar{d}(s,s')$ along with the marginalization constraint (8). This allows for tractable offline optimization by (11,16) without introducing bias, which is our novel contribution.

## E  More Discussions on LobsDICE with Finite Samples

In this section, we provide additional discussions on LobsDICE with finite samples.

### E.1  LobsDICE with sample-based approximation: distribution matching on the MLE MDP

We first show that LobsDICE that optimizes (18) is equivalent to performing optimization of (6-8) on the MLE MDP constructed by $D^I$. Let $\hat{M} \setminus R = \langle S, A, \hat{T}, \hat{p}_0, \gamma \rangle$ be the MLE MDP constructed using $D^I$. Then, constrained optimization (6-8) for the MLE MDP $\hat{M} \setminus R$ can be formulated as:

$$\max_{d,\bar{d} \geq 0} - D_{\mathrm{KL}}(\bar{d}(s,s')||\widehat{d^E}(s,s')) - \alpha D_{\mathrm{KL}}(d(s,a)||\widehat{d^I}(s,a)) \tag{46}$$

$$\text{s.t.} \sum_{a'} d(s',a') = (1-\gamma)\hat{p}_0(s') + \gamma \sum_{s,a} d(s,a)\hat{T}(s'|s,a) \quad \forall s', \tag{47}$$

$$\sum_a d(s,a)\hat{T}(s'|s,a) = \bar{d}(s,s') \quad \forall s, s', \tag{48}$$

where $\widehat{p}_0$, $\widehat{d}^E$, and $\widehat{d}^I$ are empirical distributions of $D_0$, $D^E$, and $D^I$, respectively. We consider Lagrangian for the above constrained optimization:

$$\min_{\mu,\nu} \max_{d,\bar{d}\geq 0} -\mathbb{E}_{\bar{d}}\Big[\log \frac{\bar{d}(s,s')}{d^E(s,s')}\Big] - \alpha\mathbb{E}_d\Big[\log \frac{d(s,a)}{\widehat{d}^I(s,a)}\Big] + \sum_{s,s'} \mu(s,s')\big(\bar{d}(s,s') - \sum_a d(s,a)\hat{T}(s'|s,a)\big)$$

$$+ \sum_{s'} \nu(s')\big((1-\gamma)\widehat{p}_0(s') + \gamma\sum_{s,a} d(s,a)\hat{T}(s'|s,a) - \sum_{a'} d(s',a')\big)$$

$$= \min_{\mu,\nu} \max_{d,\bar{d}\geq 0} (1-\gamma)\mathbb{E}_{s_0\sim\widehat{p}_0}[\nu(s_0)] + \mathbb{E}_{(s,s')\sim\bar{d}}\Big[\mu(s,s') - \underbrace{\log \frac{\bar{d}(s,s')}{\widehat{d}^I(s,s')}}_{=:\bar{w}(s,s')} + \underbrace{\log \frac{\widehat{d}^E(s,s')}{\widehat{d}^I(s,s')}}_{=:\widehat{r}(s,s')}\Big]$$

$$+ \mathbb{E}_{(s,a)\sim d}\Big[\underbrace{\mathbb{E}_{s'\sim\hat{T}(\cdot|s,a)}[-\mu(s,s') + \gamma\nu(s')] - \nu(s)}_{=:\widehat{e}^I_{\mu,\nu}(s,a)} -\alpha\log \underbrace{\frac{d(s,a)}{\widehat{d}^I(s,a)}}_{=:w(s,a)}\Big]$$

$$= \min_{\mu,\nu} \max_{w,\bar{w}\geq 0} (1-\gamma)\mathbb{E}_{s_0\sim\widehat{p}_0}[\nu(s_0)] + \mathbb{E}_{(s,s')\sim\widehat{d}^I}\big[\bar{w}(s,s')\big(\widehat{r}(s,s') + \mu(s,s') - \log \bar{w}(s,s')\big)\big]$$

$$+ \mathbb{E}_{(s,a)\sim\widehat{d}^I}\big[w(s,a)\big(\widehat{e}^I_{\mu,\nu}(s,a) - \alpha\log w(s,a)\big)\big] =: \mathcal{L}^I(w,\bar{w},\mu,\nu),$$

Note that this is identical to (18). We can also derive the closed-form solution $(w^I_{\mu,\nu}, \bar{w}^I_\mu)$ to the $\max_{w,\bar{w}\geq 0} \mathcal{L}^I(w,\bar{w},\mu,\nu)$:

$$w^I_{\mu,\nu}(s,a) = \exp\Big(\tfrac{1}{\alpha}\widehat{e}^I_{\mu,\nu}(s,a) - 1\Big) \;\; and \;\; \bar{w}^I_\mu(s,s') = \exp(\widehat{r}(s,s') + \mu(s,s') - 1). \qquad (49)$$

Note that this is different from (19) in that (49) is the closed-form solution for each $(s,a)$ and $(s,s')$, while (19) is the *non-parametric* closed-form solution for each sample $x \in D^I$. Exploiting (49) does *not* introduce an additional bias, but requires evaluation of the expectation within the $\exp(\cdot)$, i.e. transition model is needed.

The last step is to plug the solution $(w^I_{\mu,\nu}, \bar{w}^I_\mu)$ in (49) into $\mathcal{L}^I(w,\bar{w},\mu,\nu)$:

$$\min_{\mu,\nu} \mathcal{L}^I(w^I_{\mu,\nu}, \bar{w}^I_\mu, \mu, \nu) = (1-\gamma)\mathbb{E}_{s\sim\widehat{p}_0}[\nu(s)] + \mathbb{E}_{(s,s')\sim\widehat{d}^I}\Big[\exp\big(\widehat{r}(s,s') + \mu(s,s') - 1\big)\Big]$$

$$+ \alpha\mathbb{E}_{(s,a)\sim\widehat{d}^I}\Big[\exp\big(\tfrac{1}{\alpha}\widehat{e}^I_{\mu,\nu}(s,a) - 1\big)\Big].$$

Finally, this can be rewritten as:

$$\min_{\mu,\nu} J(\mu,\nu) = (1-\gamma)\sum_s \widehat{p}_0(s)\nu(s) + \sum_{s,s'} \widehat{d}^I(s,s')\Big[\exp\big(\widehat{r}(s,s') + \mu(s,s') - 1\big)\Big]$$

$$+ \alpha\sum_{s,a} \widehat{d}^I(s,a)\Big[\exp\Big(\tfrac{1}{\alpha}\sum_{s'} \widehat{T}^I(s'|s,a)\big(-\mu(s,s') + \gamma\nu(s') - \nu(s)\big) - 1\Big)\Big],$$

which is the objective function for our tabular LobsDICE (52).

## E.2  When will LobsDICE reduce to BCO?

LobsDICE reduces to BCO when $D^I$ is collected by an *expert* policy and $D^E$ is a dataset identical to $D^I$ except that action is missing, i.e., $D^E = \{(s,s')|(s,a,s') \in D^I\}$.

In this situation, note that $\widehat{d}^I(s,s')$ and $\widehat{d}^I(s,a)$ are valid stationary distributions on the MLE MDP constructed by $D^I$, since they satisfy both the Bellman flow constraint (47) and the marginalization constraint (48) on the MLE MDP. The $\widehat{d}^I(s,s')$ and $\widehat{d}^I(s,a)$ are also the optimal solution of (46-48), making the KL divergences to 0:

$$D_{\mathrm{KL}}(\widehat{d}^I(s,s')\|\widehat{d}^E(s,s')) = 0 \text{ and } D_{\mathrm{KL}}(\widehat{d}^I(s,a)\|\widehat{d}^I(s,a)) = 0.$$

Finally, the policy obtained by LobsDICE will simply reduce to BC on $D^I$ by noting that:

$$\pi^*(a|s) = \frac{\widehat{d}^I(s,a)}{\sum_{a'} \widehat{d}^I(s,a')}$$

Furthermore, one can easily show that BC on $D^I$ is identical to BCO (i.e. BC on $D^E$ where the missing actions are inferred by the IDM trained by $D^I$), which concludes that LobsDICE is equivalent to BCO in this special case.

However, $D^I$ will include demonstrations collected by *non-expert* policies in general, which makes the LobsDICE's solution different from (and usually better than) the BCO's solution in general cases.

# F  Theoretical Analysis

## F.1  Closed-form solutions

**Proposition 3.1.** *For any* $(\mu, \nu)$*, the closed-form solution to the inner maximization of* (11)*, i.e.* $(w_{\mu,\nu}, \bar{w}_\mu) = \arg\max_{w, \bar{w} \geq 0} \mathcal{L}(w, \bar{w}, \mu, \nu)$*, is given by:*

$$w_{\mu,\nu}(s, a) = \exp\left(\tfrac{1}{\alpha} e_{\mu,\nu}(s, a) - 1\right) \text{ and } \bar{w}_\mu(s, s') = \exp(r(s, s') + \mu(s, s') - 1). \qquad (15)$$

*Proof.* Let $e_{\mu,\nu}(s, a, s') = -\mu(s, s') + \gamma\nu(s') - \nu(s)$. Then, $e_{\mu,\nu}(s, a) = \mathbb{E}_{s' \sim T(\cdot|s,a)}[e_{\mu,\nu}(s, a, s')]$. For $(s, a)$ with $d^I(s, a) > 0$, we can compute the derivative $\frac{\partial \mathcal{L}}{\partial w(s,a)}$ of $\mathcal{L}$ w.r.t. $w(s, a)$ as follows:

$$\frac{\partial \mathcal{L}}{\partial w(s, a)} = \sum_{s'} d^I(s, a, s')(e_{\mu,\nu}(s, a, s') - \alpha \log w(s, a) - \alpha) = 0$$

$$\Leftrightarrow \sum_{s'} T(s'|s, a)(e_{\mu,\nu}(s, a, s') - \alpha \log w(s, a) - \alpha) = 0$$

$$\Leftrightarrow \sum_{s'} T(s'|s, a)(e_{\mu,\nu}(s, a, s') - \alpha) = \alpha \log w(s, a)$$

$$\Leftrightarrow w(s, a) = \exp\left(\tfrac{1}{\alpha} \mathbb{E}_{s' \sim T(\cdot|s,a)}[e_{\mu,\nu}(s, a, s')] - 1\right) = \exp\left(\tfrac{1}{\alpha} e_{\mu,\nu}(s, a) - 1\right).$$

Similar to the aforementioned derivation, when $\bar{d}^I(s, s') > 0$, we can derive the derivative of $\mathcal{L}$ w.r.t. $\bar{w}(s, s')$:

$$\frac{\partial \mathcal{L}}{\partial \bar{w}(s, s')} = \bar{d}^I(s, s')\big(\mu(s, s') - \log \bar{w}(s, s') + r(s, s') - 1\big) = 0$$

$$\Leftrightarrow \big(\mu(s, s') - \log \bar{w}(s, s') + r(s, s') - 1\big) = 0$$

$$\Leftrightarrow \log \bar{w}(s, s') = \mu(s, s') + r(s, s') - 1$$

$$\Leftrightarrow \bar{w}(s, s') = \exp\big(\mu(s, s') + r(s, s') - 1\big).$$

$\square$

**Proposition 3.2.** $\mathcal{L}(w_{\mu,\nu}, \bar{w}_\mu, \mu, \nu)$ *is convex with respect to* $\mu$ *and* $\nu$*.*

*Proof.* Using the fact that $\exp(\cdot)$ is a convex function, we can easily prove the convexity of $\mathcal{L}(w_{\mu,\nu}, \bar{w}_{\mu,\nu}, \mu, \nu)$. For brevity, let $\mathcal{L}(\mu, \nu) := \mathcal{L}(w_{\mu,\nu}, \bar{w}_{\mu,\nu}, \mu, \nu)$. Then, for given $(\mu_1, \nu_1)$, $(\mu_2, \nu_2)$ and $t \in [0, 1]$,

$$\mathcal{L}\left(t\mu_1 + (1-t)\mu_2, t\nu_1 + (1-t)\nu_2\right)$$
$$= (1-\gamma)\mathbb{E}_{p_0}[t\nu_1(s) + (1-t)\nu_2(s)] + \mathbb{E}_{d^I}[\exp(r(s, s') + t\mu_1(s, s') + (1-t)\mu_2(s, s') - 1)]$$
$$+ \alpha\mathbb{E}_{d^I}\Big[\exp\Big(\tfrac{1}{\alpha}\mathbb{E}_{s'}\big[-t\mu_1(s, s') - (1-t)\mu_2(s, s') + \gamma t\nu_1(s') + \gamma(1-t)\nu_2(s')$$
$$- t\nu_1(s) - (1-t)\nu_2(s)\big] - 1\Big)\Big]$$
$$= t(1-\gamma)\mathbb{E}_{p_0}[\nu_1(s)] + (1-t)(1-\gamma)\mathbb{E}_{p_0}[\nu_2(s)] + \mathbb{E}_{d^I}[\exp(r(s, s') + t\mu_1 + (1-t)\mu_2 - 1)]$$
$$+ \alpha\mathbb{E}_{d^I}\Big[\exp\Big(\tfrac{1}{\alpha}\mathbb{E}_{s'}\big[-t\mu_1(s, s') - (1-t)\mu_2(s, s') + \gamma t\nu_1(s') + \gamma(1-t)\nu_2(s')$$
$$- t\nu_1(s) - (1-t)\nu_2(s)\big] - 1\Big)\Big]$$

$$(Cont.) \leq t(1-\gamma)\mathbb{E}_{p_0}[\nu_1(s)] + (1-t)(1-\gamma)\mathbb{E}_{p_0}[\nu_2(s)]$$
$$+ t\mathbb{E}_{d^I}[\exp(r(s,s') + \mu_1(s,s') - 1)] + (1-t)\mathbb{E}_{d^I}[\exp(r(s,s') + \mu_2(s,s') - 1)]$$
$$+ \alpha\mathbb{E}_{d^I}\Big[\exp\Big(\tfrac{1}{\alpha}\mathbb{E}_{s'}\big[-t\mu_1(s,s') - (1-t)\mu_2(s,s') + \gamma t\nu_1(s') + \gamma(1-t)\nu_2(s')$$
$$- t\nu_1(s) - (1-t)\nu_2(s)\big] - 1\Big)\Big]$$
$$\leq t(1-\gamma)\mathbb{E}_{p_0}[\nu_1(s)] + (1-t)(1-\gamma)\mathbb{E}_{p_0}[\nu_2(s)]$$
$$+ t\mathbb{E}_{d^I}[\exp(r(s,s') + \mu_1(s,s') - 1)] + (1-t)\mathbb{E}_{d^I}[\exp(r(s,s') + \mu_2(s,s') - 1)]$$
$$+ t\alpha\mathbb{E}_{d^I}\Big[\exp\Big(\tfrac{1}{\alpha}\mathbb{E}_{s'}\big[\mu_1(s,s') + \gamma\nu_1(s') - \nu_1(s)\big] - 1\Big)\Big]$$
$$+ (1-t)\alpha\mathbb{E}_{d^I}\Big[\exp\Big(\tfrac{1}{\alpha}\mathbb{E}_{s'}\big[-\mu_2(s,s') + \gamma\nu_2(s') - \nu_2(s)\big] - 1\Big)\Big]$$
$$= t\mathcal{L}(\mu_1,\nu_1) + (1-t)\mathcal{L}(\mu_2,\nu_2).$$

For the inequalities in the above formulation, we use the fact that $\mathbb{E}_{d^I}[\exp(\cdot)]$ is a instance of convex functions. $\qquad\square$

**Proposition 3.3.** *For any $\nu$, the closed-form solution to the minimization* (20) *with respect to $\mu$, i.e.* $\mu_\nu = \arg\min_\mu \widehat{\mathcal{L}}(\mu,\nu)$, *is*

$$\mu_\nu(s,s') = \tfrac{1}{1+\alpha}\big(-\alpha r(s,s') + \gamma\nu(s') - \nu(s)\big). \tag{21}$$

*Proof.* For simplicity, let $\widehat{\mathcal{L}}(\mu,\nu) := \widehat{\mathcal{L}}(w_{\mu,\nu}, \bar{w}_{\mu,\nu}, \mu, \nu)$ and $x := (s,a,s')$. Then,

$$\tfrac{\partial\widehat{\mathcal{L}}}{\partial\mu(x)} = d^I(x)\Big[\exp(r(x) + \mu(x) - 1) - \exp\Big(\tfrac{1}{\alpha}(-\mu(x) + \gamma\nu(s') - \nu(s)) - 1\Big)\Big] = 0$$
$$\Leftrightarrow \exp(r(x) + \mu(x) - 1) = \exp\Big(\tfrac{1}{\alpha}(-\mu(x) + \gamma\nu(s') - \nu(s)) - 1\Big)$$
$$\Leftrightarrow \alpha(r(x) + \mu(x)) = -\mu(x) + \gamma\nu(s') - \nu(s)$$
$$\Leftrightarrow (1+\alpha)\mu(x) = -\alpha r(x) + \gamma\nu(s') - \nu(s)$$
$$\Leftrightarrow \mu(x) = \tfrac{1}{1+\alpha}\big(-\alpha r(x) + \gamma\nu(s') - \nu(s)\big)$$

$\qquad\square$

## F.2 Surrogate Objective

We first show a property of $\tilde{\mathcal{L}}$ that will be used to prove propositions:

**Lemma F.1.** *For given function $\nu : S \to \mathbb{R}$,*

$$\tilde{\mathcal{L}}(\nu) = \tilde{\mathcal{L}}(\nu + C) \quad \forall C \in \mathbb{R}.$$

*Proof.* For any $C \in \mathbb{R}$,

$\tilde{\mathcal{L}}(\nu + C)$

$$= (1-\gamma)\mathbb{E}_{s\sim p_0}[\nu(s) + C] + (1+\alpha)\log\mathbb{E}_{(s,a,s')\sim d^I}\Big[\exp\Big(\tfrac{1}{1+\alpha}\hat{A}_{\nu+C}(s,a,s')\Big)\Big]$$
$$= (1-\gamma)\mathbb{E}_{s\sim p_0}[\nu(s) + C] + (1+\alpha)\log\mathbb{E}_{(s,a,s')\sim d^I}\Big[\exp\Big(\tfrac{1}{1+\alpha}\big(r(s,s') + \gamma(\nu(s') + C) - (\nu(s) + C)\big)\Big)\Big]$$
$$= (1-\gamma)\mathbb{E}_{s\sim p_0}[\nu(s) + C] + (1+\alpha)\log\mathbb{E}_{(s,a,s')\sim d^I}\Big[\exp\Big(\tfrac{1}{1+\alpha}\big(r(s,s') + \gamma\nu(s') - \nu(s)\big)\Big)\exp\Big(\tfrac{(\gamma-1)C}{1+\alpha}\Big)\Big]$$
$$= (1-\gamma)\mathbb{E}_{s\sim p_0}[\nu(s) + C] + (1+\alpha)\log\mathbb{E}_{(s,a,s')\sim d^I}\Big[\exp\Big(\tfrac{1}{1+\alpha}(r(s,s') + \gamma\nu(s') - \nu(s))\Big)\Big] + (\gamma-1)C$$
$$= (1-\gamma)\mathbb{E}_{s\sim p_0}[\nu(s)] + (1+\alpha)\log\mathbb{E}_{(s,a,s')\sim d^I}\Big[\exp\Big(\tfrac{1}{1+\alpha}(r(s,s') + \gamma\nu(s') - \nu(s))\Big)\Big]$$
$$= \tilde{\mathcal{L}}(\nu).$$

$\qquad\square$

Now, we prove Proposition 3.4 and Proposition 3.5:

**Proposition 3.4.** *Let $\widetilde{\mathcal{L}}(\widetilde{\nu})$ be the function:*

$$\widetilde{\mathcal{L}}(\widetilde{\nu}) = (1-\gamma)\hat{\mathbf{E}}_{s_0 \in D_0}[\widetilde{\nu}(s_0)] + (1+\alpha)\log\hat{\mathbf{E}}_{x \in D^I}\left[\exp\left(\tfrac{1}{1+\alpha}\widehat{A}_{\widetilde{\nu}}(s,a,s')\right)\right]. \qquad (23)$$

*Then, $\min_{\widehat{\nu}} \widehat{\mathcal{L}}(\widehat{\nu}) = \min_{\widetilde{\nu}} \widetilde{\mathcal{L}}(\widetilde{\nu})$ holds. Also, $\widetilde{\mathcal{L}}(\widetilde{\nu})$ is convex with respect to $\widetilde{\nu}$.*

*Proof.* First of all, from the fact that $\log(x+1) \leq x$ for all $x > -1$, we can easily conclude that $\min_{\nu} \tilde{\mathcal{L}}(\nu) \leq \min_{\nu} \hat{\mathcal{L}}(\nu)$. Now, we will show that $\min_{\nu} \hat{\mathcal{L}}(\nu) \leq \min_{\nu} \tilde{\mathcal{L}}(\nu)$. For given $\nu^* \in \arg\min_{\nu} \tilde{\mathcal{L}}(\nu)$, we define a constant $C$ as follows:

$$C = \tfrac{1+\alpha}{1-\gamma}\log\mathbb{E}_{(s,a,s')\sim d^I}\left[\exp\left(\tfrac{1}{1+\alpha}\hat{A}_{\nu^*}(s,a,s')-1\right)\right],$$

which implies

$$\exp\left(\tfrac{(1-\gamma)C}{1+\alpha}\right) = \mathbb{E}_{(s,a,s')\sim d^I}\left[\exp\left(\tfrac{1}{1+\alpha}\hat{A}_{\nu^*}(s,a,s')-1\right)\right].$$

Then, from the following two equations:

$\hat{\mathcal{L}}(\nu^* + C)$
$$= (1-\gamma)\mathbb{E}_{s\sim p_0}[\nu^*(s)+C] + (1+\alpha)\mathbb{E}_{(s,a,s')\sim d^I}\left[\exp\left(\tfrac{1}{1+\alpha}\hat{A}_{\nu^*+C}(s,a,s')-1\right)\right]$$
$$= (1-\gamma)\mathbb{E}_{s\sim p_0}[\nu^*(s)+C] + (1+\alpha)\mathbb{E}_{(s,a,s')\sim d^I}\left[\exp\left(\tfrac{1}{1+\alpha}\hat{A}_{\nu^*}(s,a,s')-1\right)\exp\left(\tfrac{(\gamma-1)C}{1+\alpha}\right)\right]$$
$$= (1-\gamma)\mathbb{E}_{s\sim p_0}[\nu^*(s)+C] + (1+\alpha)\mathbb{E}_{(s,a,s')\sim d^I}\left[\exp\left(\tfrac{1}{1+\alpha}\hat{A}_{\nu^*}(s,a,s')-1\right)\right]\exp\left(-\tfrac{(1-\gamma)C}{1+\alpha}\right)$$
$$= (1-\gamma)\mathbb{E}_{s\sim p_0}[\nu^*(s)+C] + (1+\alpha)$$

and

$\tilde{\mathcal{L}}(\nu^* + C)$
$$= (1-\gamma)\mathbb{E}_{s\sim p_0}[\nu^*(s)+C] + (1+\alpha)\log\mathbb{E}_{(s,a,s')\sim d^I}\left[\exp\left(\tfrac{1}{1+\alpha}\hat{A}_{\nu^*+C}(s,a,s')\right)\right]$$
$$= (1-\gamma)\mathbb{E}_{s\sim p_0}[\nu^*(s)+C] + (1+\alpha)\log\mathbb{E}_{(s,a,s')\sim d^I}\left[\exp\left(\tfrac{1}{1+\alpha}\hat{A}_{\nu^*}(s,a,s')\right)\right] + (\gamma-1)C$$
$$= (1-\gamma)\mathbb{E}_{s\sim p_0}[\nu^*(s)+C] + (1+\alpha)\log\mathbb{E}_{(s,a,s')\sim d^I}\left[\exp\left(\tfrac{1}{1+\alpha}\hat{A}_{\nu^*}(s,a,s')\right)\right]$$
$$\quad - (1+\alpha)\log\mathbb{E}_{(s,a,s')\sim d^I}\left[\exp\left(\tfrac{1}{1+\alpha}\hat{A}_{\nu^*}(s,a,s')-1\right)\right]$$
$$= (1-\gamma)\mathbb{E}_{s\sim p_0}[\nu^*(s)+C] + (1+\alpha)\log\mathbb{E}_{(s,a,s')\sim d^I}\left[\exp\left(\tfrac{1}{1+\alpha}\hat{A}_{\nu^*}(s,a,s')\right)\right]$$
$$\quad - (1+\alpha)\left(\log\mathbb{E}_{(s,a,s')\sim d^I}\left[\exp\left(\tfrac{1}{1+\alpha}\hat{A}_{\nu^*}(s,a,s')\right)\right]-1\right)$$
$$= (1-\gamma)\mathbb{E}_{s\sim p_0}[\nu^*(s)+C] + 1 + \alpha,$$

we can conclude that

$$\hat{\mathcal{L}}(\nu^* + C) = \tilde{\mathcal{L}}(\nu^* + C).$$

From the Lemma F.1, we obtain

$$\min_{\nu} \hat{\mathcal{L}}(\nu) \leq \hat{\mathcal{L}}(\nu^* + C) = \tilde{\mathcal{L}}(\nu^* + C) = \min_{\nu} \tilde{\mathcal{L}}(\nu).$$

We show that $\min_{\nu} \tilde{\mathcal{L}}(\nu) \leq \min_{\nu} \hat{\mathcal{L}}(\nu)$ and $\min_{\nu} \hat{\mathcal{L}}(\nu) \leq \min_{\nu} \tilde{\mathcal{L}}(\nu)$, so $\min_{\nu} \hat{\mathcal{L}}(\nu) = \min_{\nu} \tilde{\mathcal{L}}(\nu)$.

Finally, similar to the proof steps for Proposition 3.2, we can easily show that $\hat{\mathcal{L}}_{\mathrm{FD}}(\nu)$ is convex w.r.t. $\nu$ (Remark that log-sum-exp is a convex function). $\qquad\square$

**Proposition 3.5.** *Let $\widehat{V}$ and $\widetilde{V}$ be the sets $\arg\min_{\widehat{\nu}} \widehat{\mathcal{L}}(\widehat{\nu})$ and $\arg\min_{\widetilde{\nu}} \widetilde{\mathcal{L}}(\widetilde{\nu})$, respectively. Then, $\widetilde{V} = \{\widehat{\nu}^* + C \mid \widehat{\nu}^* \in \widehat{V}, C \in \mathbb{R}\}$ holds.*

*Proof.* We will prove this proposition by showing $\tilde{V} \subseteq \{\nu^* + C' | \nu^* \in \hat{V}, C \in \mathbb{R}\}$ and $\tilde{V} \supseteq \{\nu^* + C' | \nu^* \in \hat{V}, C \in \mathbb{R}\}$.

($\subseteq$) For given $\nu^* \in \arg\min_\nu \tilde{\mathcal{L}}(\nu)$, we define a constant $C$ as follows:

$$C = \tfrac{1+\alpha}{1-\gamma} \log \mathbb{E}_{(s,a,s')\sim d^I} \Big[ \exp \big( \tfrac{1}{1+\alpha} \hat{A}_{\nu^*}(s,a,s') - 1 \big) \Big],$$

which implies

$$\exp \Big( \tfrac{(1-\gamma)C}{1+\alpha} \Big) = \mathbb{E}_{(s,a,s')\sim d^I} \Big[ \exp \big( \tfrac{1}{1+\alpha} \hat{A}_{\nu^*}(s,a,s') - 1 \big) \Big].$$

Then, by the proof steps of Proposition 3.4, we obtain

$$\hat{\mathcal{L}}(\nu^* + C) = \tilde{\mathcal{L}}(\nu^* + C) = \min_\nu \tilde{\mathcal{L}}(\nu) = \min_\nu \hat{\mathcal{L}}(\nu).$$

It means $(\nu^* + C) \in \arg\min_\nu \hat{\mathcal{L}}(\nu)$ and thus,

$$\nu^* \in \{\hat{\nu}^* + C | \hat{\nu}^* \in \arg\min_\nu \hat{\mathcal{L}}(\nu), C \in \mathbb{R}\},$$

i.e.,

$$\arg\min_\nu \tilde{\mathcal{L}}(\nu) \subseteq \{\hat{\nu}^* + C | \hat{\nu}^* \in \arg\min_\nu \hat{\mathcal{L}}(\nu), C \in \mathbb{R}\}$$

($\supseteq$) Let $\hat{\nu}^* \in \arg\min_\nu \hat{\mathcal{L}}(\nu)$. Then,

$$\frac{d^*(s,a)}{d^I(s,a)} = w^*_{\mu^*,\nu^*}(s,a) = \exp \Big( \tfrac{1}{\alpha} \big( \mathbb{E}_{s'\sim T(\cdot|s,a)}[\gamma\nu(s') - \mu(s,s')] - \nu(s) \big) - 1 \Big),$$

$$\frac{d^*(s,s')}{d^I(s,s')} = \bar{w}^*_{\mu^*,\nu^*}(s,s') = \exp(r(s,s') + \mu^*(s,s') - 1).$$

Let $C \in \mathbb{R}$. Then, we derive the following equation:

$$\tilde{\mathcal{L}}(\hat{\nu}^* + C) = \tilde{\mathcal{L}}(\hat{\nu}^*)$$
$$= (1-\gamma)\mathbb{E}_{s\sim p_0}[\hat{\nu}^*(s)] + (1+\alpha) \log \mathbb{E}_{(s,a,s')\sim d^I} \Big[ \exp \big( \tfrac{1}{1+\alpha} \hat{A}_{\hat{\nu}^*}(s,a,s') \big) \Big]$$
$$= (1-\gamma)\mathbb{E}_{s\sim p_0}[\hat{\nu}^*(s)] + (1+\alpha) \log \mathbb{E}_{(s,a,s')\sim d^I} \Big[ \exp \big( \tfrac{1}{1+\alpha} \hat{A}_{\hat{\nu}^*}(s,a,s') - 1 + 1 \big) \Big]$$
$$= (1-\gamma)\mathbb{E}_{s\sim p_0}[\nu^*(s)] + 1 + \alpha.$$

Here, we apply Lemma F.1 to derive the first equality. Because

$$\min_\nu \tilde{\mathcal{L}}(\nu) = \min_\nu \hat{\mathcal{L}}(\nu)$$
$$= \hat{\mathcal{L}}(\hat{\nu}^*)$$
$$= (1-\gamma)\mathbb{E}_{s\sim p_0}[\hat{\nu}^*(s)] + (1+\alpha)\mathbb{E}_{(s,a,s')\sim d^I} \Big[ \exp \big( \tfrac{1}{1+\alpha} \hat{A}_{\hat{\nu}^*}(s,a,s') - 1 \big) \Big]$$
$$= (1-\gamma)\mathbb{E}_{s\sim p_0}[\nu^*(s)] + 1 + \alpha,$$

we can conclude that $\mathcal{L}_{\mathrm{FD}}(\hat{\nu}^* + C) = \min_\nu \mathcal{L}_{\mathrm{FD}}(\nu)$, i.e., $(\hat{\nu}^* + C) \in \arg\min_\nu \tilde{\mathcal{L}}(\nu)$. Consequently,

$$\arg\min_\nu \tilde{\mathcal{L}}(\nu) \supseteq \{\hat{\nu}^* + C | \hat{\nu}^* \in \arg\min_\nu \hat{\mathcal{L}}(\nu), C \in \mathbb{R}\}.$$

$\square$

### F.3 Fenchel dual formulation

Let

$$\delta_C(x) := \begin{cases} 0 & x \in C \\ \infty & \text{otherwise} \end{cases}.$$

Then we can provide following proposition:

**Proposition F.2.** *We can rewrite the optimization problem (6-8) as*

$$\max_{d \geq 0} - \delta_{(1-\gamma)p_0}\left(-(\gamma\mathcal{T} - \mathcal{B})_* d\right) - D_{\text{KL}}\left((\bar{\mathcal{T}}_* d)\|\bar{d}^E\right) - \alpha D_{\text{KL}}(d\|d^I). \tag{50}$$

*Then, the dual problem of* (50) *is given by*

$$\min_{\mu,\nu} \mathcal{L}_{FD}(\mu,\nu) \tag{51}$$

$$:= (1-\gamma)\mathbb{E}_{s\sim p_0}[\nu(s)] + \log \mathbb{E}_{\bar{d}^I}\left[\exp\left(r(s,s') + \mu(s,s')\right)\right] + \alpha \log \mathbb{E}_{d^I}\left[\exp\left(\frac{1}{\alpha}e_{\mu,\nu}(s,a)\right)\right].$$

*Proof.* We first define following three functions

$$f(\cdot) := \delta_{\{(1-\gamma)p_0\}}(\cdot),$$
$$g(\cdot;r) := \langle\cdot, -r\rangle + D_{\text{KL}}(\cdot\|\bar{\mathcal{T}}_* d^I),$$
$$h(\cdot;\mu) := \langle\cdot, \bar{\mathcal{T}}\mu\rangle + \alpha D_{\text{KL}}(\cdot\|d^I),$$

and conjugate functions,

$$f_*(\cdot) := (1-\gamma)\mathbb{E}_{s\sim p_0}[\cdot],$$
$$g_*(\cdot;r) := \log \mathbb{E}_{(s,s')\sim\bar{\mathcal{T}}_* d^I}[\exp(\cdot + r(s,s'))],$$
$$h_*(\cdot;\mu) := \alpha \log \mathbb{E}_{(s,a)\sim d^I}\left[\exp\left(\frac{\cdot - (\bar{\mathcal{T}}\mu)(s,a)}{\alpha}\right)\right].$$

Then, the dual formulation of the primal (50) can be derived as follows:

$$\max_{d\geq 0} - \delta_{(1-\gamma)p_0}\left(-(\gamma\mathcal{T}-\mathcal{B})_* d\right) - \mathbb{E}_{\substack{(s,a)\sim d,\\ s'\sim T(\cdot|s,a)}}\left[\log\frac{(\bar{\mathcal{T}}_* d)(s,s')}{d^E(s,s')}\right] - \alpha D_{\text{KL}}(d\|d^I)$$

$$= \max_{d\geq 0} - f\left(-(\gamma\mathcal{T}-\mathcal{B})_* d\right) + \mathbb{E}_{\substack{(s,a)\sim d,\\ s'\sim T(\cdot|s,a)}}\left[-\log\frac{(\bar{\mathcal{T}}_* d)(s,s')}{(\bar{\mathcal{T}}_* d^I)(s,s')} + \underbrace{\log\frac{d^E(s,s')}{(\bar{\mathcal{T}}_* d^I)(s,s')}}_{:=r(s,s')}\right] - \alpha D_{\text{KL}}(d\|d^I)$$

$$= \max_{d\geq 0} - f\left(-(\gamma\mathcal{T}-\mathcal{B})_* d\right) - g(\bar{\mathcal{T}}_* d;r) - \alpha D_{\text{KL}}(d\|d^I)$$

$$= \max_{d\geq 0} - f\left(-(\gamma\mathcal{T}-\mathcal{B})_* d\right) - \left\{\max_\mu -g_*(\mu;r) + \langle\bar{\mathcal{T}}_* d,\mu\rangle\right\} - \alpha D_{\text{KL}}(d\|d^I)$$

$$= \max_{d\geq 0}\min_\mu - f\left(-(\gamma\mathcal{T}-\mathcal{B})_* d\right) + g_*(\mu;r) - \langle\bar{\mathcal{T}}_* d,\mu\rangle - \alpha D_{\text{KL}}(d\|d^I)$$

$$= \max_{d\geq 0}\min_\mu - f\left(-(\gamma\mathcal{T}-\mathcal{B})_* d\right) + g_*(\mu;r) - \langle d,\bar{\mathcal{T}}\mu\rangle - \alpha D_{\text{KL}}(d\|d^I)$$

$$= \max_{d\geq 0}\min_\mu - f\left(-(\gamma\mathcal{T}-\mathcal{B})_* d\right) + g_*(\mu;r) - h(d;\mu)$$

$$= \max_{d\geq 0}\min_\mu - \left\{\max_\nu\langle-(\gamma\mathcal{T}-\mathcal{B})_* d,\nu\rangle - f_*(\nu)\right\} + g_*(\mu;r) - h(d;\mu)$$

$$= \max_{d\geq 0}\min_{\nu,\mu}\langle(\gamma\mathcal{T}-\mathcal{B})_* d,\nu\rangle + f_*(\nu) + g_*(\mu;r) - h(d;\mu)$$

$$= \max_{d\geq 0}\min_{\nu,\mu}\langle d,(\gamma\mathcal{T}-\mathcal{B})\nu\rangle + f_*(\nu) + g_*(\mu;r) - h(d;\mu) := \mathcal{L}_{FD}(d;\mu,\nu)$$

Here, we can reorder the maximin to minimax and therefore, we can derive the

$$\min_{\nu,\mu}\max_{d\geq 0}\mathcal{L}_{FD}(d;\mu,\nu)$$

$$= \min_{\nu,\mu}\max_{d\geq 0}\langle d,(\gamma\mathcal{T}-\mathcal{B})\nu\rangle + f_*(\nu) + g_*(\mu;r) - h(d;\mu)$$

$$= \min_{\nu,\mu}\left\{\max_{d\geq 0}\langle d,(\gamma\mathcal{T}-\mathcal{B})\nu\rangle - h(d;\mu)\right\} + f_*(\nu) + g_*(\mu;r)$$

$$= \min_{\nu,\mu}h_*((\gamma\mathcal{T}-\mathcal{B})\nu) + f_*(\nu) + g_*(\mu;r).$$

We can rewrite the last term as
$$\min_{\nu,\mu}(1-\gamma)\mathbb{E}_{s\sim p_0}[\nu(s)] + \log \mathbb{E}_{\substack{(s,a)\sim d^I,\\ s'\sim T(\cdot|s,a)}}[\exp(\mu(s,s')+r(s,s'))] + \alpha \log \mathbb{E}_{(s,a)\sim d^I}\Big[\exp\Big(\tfrac{1}{\alpha}e_{\mu,\nu}(s,a)\Big)\Big].$$

$\square$

**Lemma F.3.** *For given function* $\mu : S \times S \to \mathbb{R}$ *and* $\nu : S \to \mathbb{R}$,
$$\mathcal{L}_{FD}(\mu,\nu) = \mathcal{L}_{FD}(\mu + C, \nu + C') \quad \forall C, C' \in \mathbb{R}.$$

*Proof.*

$\mathcal{L}_{\text{FD}}(\mu + C, \nu + C')$

$= (1-\gamma)\mathbb{E}_{s\sim p_0}[\nu(s)+C'] + \log \mathbb{E}_{(s,a,s')\sim d^I}[\exp(\mu(s,s')+C+r(s,s'))]$

$\quad + \alpha \log \mathbb{E}_{(s,a)\sim d^I}\Big[\exp\Big(\tfrac{1}{\alpha}\big(\mathbb{E}_{s'\sim T(\cdot|s,a)}[\gamma\nu(s')+\gamma C'-\mu(s,s')-C]-\nu(s)-C'\big)\Big)\Big]$

$= (1-\gamma)\mathbb{E}_{s\sim p_0}[\nu(s)] + (1-\gamma)C' + \log \mathbb{E}_{(s,a,s')\sim d^I}[\exp(\mu(s,s')+r(s,s'))]+C$

$\quad + \alpha \log \mathbb{E}_{(s,a)\sim d^I}\Big[\exp\Big(\tfrac{1}{\alpha}\big(\mathbb{E}_{s'\sim T(\cdot|s,a)}[\gamma\nu(s')-\mu(s,s')]-\nu(s)\big)\Big)\Big] + (\gamma C'-C-C')$

$= (1-\gamma)\mathbb{E}_{s\sim p_0}[\nu(s)] + \log \mathbb{E}_{(s,a,s')\sim d^I}[\exp(\mu(s,s')+r(s,s'))]$

$\quad + \alpha \log \mathbb{E}_{(s,a)\sim d^I}\Big[\exp\Big(\tfrac{1}{\alpha}\big(\mathbb{E}_{s'\sim T(\cdot|s,a)}[\gamma\nu(s')-\mu(s,s')]-\nu(s)\big)\Big)\Big]$

$= \mathcal{L}_{\text{FD}}(\mu,\nu).$

$\square$

Finally, we can show that the relation between $\arg\min_{\mu,\nu}\mathcal{L}$ and $\arg\min_{\mu,\nu}\mathcal{L}_{\text{FD}}$:

**Proposition F.4.** *Let* $V$ *and* $V_{FD}$ *be the set of optimal solutions* $(\mu^*,\nu^*)$ *of* $\arg\min_{\mu,\nu}\mathcal{L}(w_{\nu,\mu},\bar{w}_{\nu,\mu},\mu,\nu)$ *and* $\arg\min_{\mu,\nu}\mathcal{L}_{FD}(\mu,\nu)$, *respectively. Then,*
$$V_{FD} = \{(\mu^* + C, \nu^* + C')|(\mu^*,\nu^*) \in V, C \in \mathbb{R}, C' \in \mathbb{R}\}$$
*holds.*

*Proof.* For brevity, we will denote $\mathcal{L}(w_{\nu,\mu},\bar{w}_{\nu,\mu},\mu,\nu)$ as $\mathcal{L}(\mu,\nu)$.

($\subseteq$) For given $(\hat{\mu}^*,\hat{\nu}^*) \in \arg\min_{\mu,\nu}\mathcal{L}_{\text{FD}}(\mu,\nu)$, we define two constants as follows:

$\quad C = -\log \mathbb{E}_{(s,a,s')\sim d^I}[\exp(\hat{\mu}^*(s,s')+r(s,s')-1)],$

$\quad C' = \dfrac{\alpha}{1-\gamma}\log \mathbb{E}_{(s,a)\sim d^I}\Big[\exp\Big(\dfrac{\mathbb{E}_{s'}[\gamma\hat{\nu}^*(s')-\hat{\mu}^*(s,s')]-\hat{\nu}^*(s')}{\alpha}-1\Big)\Big] - \dfrac{C}{1-\gamma}.$

Then, from the following two equations:

$\mathcal{L}(\hat{\mu}^* + C, \hat{\nu}^* + C')$

$= (1-\gamma)\mathbb{E}_{s\sim p_0}[\hat{\nu}^*(s)+C'] + \mathbb{E}_{(s,a,s')\sim d^I}[\exp(\hat{\mu}^*(s,s')+C+r(s,s')-1)]$

$\quad + \alpha\mathbb{E}_{(s,a)\sim d^I}\Big[\exp\Big(\tfrac{1}{\alpha}\big(\mathbb{E}_{s'}[\gamma\hat{\nu}^*(s')+\gamma C'-\hat{\mu}^*(s,s')-C]-\hat{\nu}^*(s')-C'\big)-1\Big)\Big]$

$= (1-\gamma)\mathbb{E}_{s\sim p_0}[\hat{\nu}^*(s)] + (1-\gamma)C' + \mathbb{E}_{(s,a,s')\sim d^I}[\exp(\hat{\mu}^*(s,s')+r(s,s')-1)]\exp(C)$

$\quad + \alpha\mathbb{E}_{(s,a)\sim d^I}\Big[\exp\Big(\tfrac{1}{\alpha}\big(\mathbb{E}_{s'}[\gamma\hat{\nu}^*(s')-\hat{\mu}^*(s,s')]-\hat{\nu}^*(s')\big)-1\Big)\Big]\exp\Big(\tfrac{1}{\alpha}\big(-(1-\gamma)C'-C\big)\Big)$

$= (1-\gamma)\mathbb{E}_{s\sim p_0}[\hat{\nu}^*(s)] + (1-\gamma)C' + 1 + \alpha,$

and

$\mathcal{L}_{\text{FD}}(\hat{\mu}^* + C, \hat{\nu}^* + C')$

$\quad = (1-\gamma)\mathbb{E}_{s\sim p_0}[\hat{\nu}^*(s)+C'] + \log \mathbb{E}_{(s,a,s')\sim d^I}[\exp(\hat{\mu}^*(s,s')+C+r(s,s'))]$

$\quad\quad + \alpha \log \mathbb{E}_{(s,a)\sim d^I}\Big[\exp\Big(\tfrac{1}{\alpha}\big(\mathbb{E}_{s'}[\gamma\hat{\nu}^*(s')+\gamma C'-\hat{\mu}^*(s,s')-C]-\hat{\nu}^*(s')-C'\big)\Big)\Big]$

$\quad = (1-\gamma)\mathbb{E}_{s\sim p_0}[\hat{\nu}^*(s)] + (1-\gamma)C' + \log \mathbb{E}_{(s,a,s')\sim d^I}[\exp(\hat{\mu}^*(s,s')+r(s,s'))]+C$

$\quad\quad + \alpha \log \mathbb{E}_{(s,a)\sim d^I}\Big[\exp\Big(\tfrac{1}{\alpha}\big(\mathbb{E}_{s'}[\gamma\hat{\nu}^*(s')-\hat{\mu}^*(s,s')]-\hat{\nu}^*(s')\big)\Big)\Big] + (\gamma C'-C-C')$

$\quad = (1-\gamma)\mathbb{E}_{s\sim p_0}[\hat{\nu}^*(s)+C'] + 1 + \alpha,$

we can conclude that

$$\mathcal{L}(\hat{\mu}^* + C, \hat{\nu}^* + C') = \mathcal{L}_{\text{FD}}(\hat{\mu}^* + C, \hat{\nu}^* + C') = \mathcal{L}_{\text{FD}}(\hat{\mu}^*, \hat{\nu}^*) = \min_{\mu,\nu} \mathcal{L}_{\text{FD}}(\mu,\nu) = \min_{\mu,\nu} \mathcal{L}(\mu,\nu).$$

It means $(\hat{\mu}^* + C, \hat{\nu}^* + C') \in \arg\min_{\mu,\nu} \mathcal{L}(\mu,\nu)$ and thus,

$$(\hat{\mu}^*, \hat{\nu}^*) \in \{\mu + C, \nu + C' | (\mu,\nu) \in \arg\min_{\mu,\nu} \mathcal{L}(\mu,\nu), C \in \mathbb{R}, C' \in \mathbb{R}\},$$

i.e.,

$$\arg\min_{\mu,\nu} \mathcal{L}_{\text{FD}}(\mu,\nu) \subseteq \{\mu + C, \nu + C' | (\mu,\nu) \in \arg\min_{\mu,\nu} \mathcal{L}(\mu,\nu), C \in \mathbb{R}, C' \in \mathbb{R}\}$$

($\supseteq$) Let $(\mu^*, \nu^*) \in \arg\min_{\mu,\nu} \mathcal{L}(\mu,\nu)$. Then,

$$\frac{d^*(s,a)}{d^I(s,a)} = w^*_{\mu^*,\nu^*}(s,a) = \exp\left(\frac{1}{\alpha}\left(\mathbb{E}_{s' \sim T(\cdot|s,a)}[\gamma\nu(s') - \mu(s,s')] - \nu(s)\right) - 1\right),$$

$$\frac{d^*(s,s')}{d^I(s,s')} = \bar{w}^*_{\mu^*,\nu^*}(s,s') = \exp(r(s,s') + \mu^*(s,s') - 1).$$

Let $C \in \mathbb{R}$ and $C' \in \mathbb{R}$. Then, we derive the following equation:

$$\mathcal{L}_{\text{FD}}(\mu^* + C, \nu^* + C')$$
$$= \mathcal{L}_{\text{FD}}(\mu^*, \nu^*)$$
$$= (1-\gamma)\mathbb{E}_{s \sim p_0}[\nu^*(s)] + \log\mathbb{E}_{(s,a,s') \sim d^I}[\exp(\mu^*(s,s') + r(s,s'))]$$
$$\quad + \alpha\log\mathbb{E}_{(s,a) \sim d^I}\left[\exp\left(\frac{1}{\alpha}\left(\mathbb{E}_{s' \sim T(\cdot|s,a)}[\gamma\nu^*(s') - \mu^*(s,s')] - \nu^*(s)\right)\right)\right]$$
$$= (1-\gamma)\mathbb{E}_{s \sim p_0}[\nu^*(s)] + \log\mathbb{E}_{(s,a,s') \sim d^I}[\bar{w}^*_{\mu^*,\nu^*}(s,s')\exp(1)]$$
$$\quad + \alpha\log\mathbb{E}_{(s,a) \sim d^I}[w(s,a)\exp(1)]$$
$$= (1-\gamma)\mathbb{E}_{s \sim p_0}[\nu^*(s)] + \log\mathbb{E}_{(s,a,s') \sim d^*}[\exp(1)] + \alpha\log\mathbb{E}_{(s,a) \sim d^*}[\exp(1)]$$
$$= (1-\gamma)\mathbb{E}_{s \sim p_0}[\nu^*(s)] + 1 + \alpha$$
$$= (1-\gamma)\mathbb{E}_{s \sim p_0}[\nu^*(s)] + 1 + \alpha.$$

Here, we apply Lemma F.1 to derive the first equality. Because

$$\min_{\mu,\nu} \mathcal{L}_{\text{FD}}(\mu,\nu)$$
$$= \min_{\mu,\nu} \mathcal{L}(\mu,\nu)$$
$$= \mathcal{L}(\mu^*, \nu^*)$$
$$= (1-\gamma)\mathbb{E}_{s \sim p_0}[\nu^*(s)] + \mathbb{E}_{(s,a,s') \sim d^I}[\exp(\mu^*(s,s') + r(s,s') - 1)]$$
$$\quad + \alpha\mathbb{E}_{(s,a) \sim d^I}\left[\exp\left(\frac{1}{\alpha}\left(\mathbb{E}_{s' \sim T(\cdot|s,a)}[\gamma\nu^*(s') - \mu^*(s,s')] - \nu^*(s)\right) - 1\right)\right]$$
$$= (1-\gamma)\mathbb{E}_{s \sim p_0}[\nu^*(s)] + \mathbb{E}_{(s,a,s') \sim d^I}[\bar{w}^*_{\mu^*,\nu^*}(s,s')] + \alpha\mathbb{E}_{(s,a) \sim d^I}[w^*_{\mu^*,\nu^*}(s,a)]$$
$$= (1-\gamma)\mathbb{E}_{s \sim p_0}[\nu^*(s)] + \mathbb{E}_{(s,a,s') \sim d^*}[1] + \alpha\mathbb{E}_{(s,a) \sim d^*}[1]$$
$$= (1-\gamma)\mathbb{E}_{s \sim p_0}[\nu^*(s)] + 1 + \alpha,$$

we can conclude that $\mathcal{L}_{\text{FD}}(\mu^* + C, \nu^* + C') = \min_{\mu,\nu} \mathcal{L}_{\text{FD}}(\mu,\nu)$, i.e., $(\mu^* + C, \nu^* + C') \in \arg\min_{\mu,\nu} \mathcal{L}_{\text{FD}}(\mu,\nu)$. It means,

$$\arg\min_{\mu,\nu} \mathcal{L}_{\text{FD}}(\mu,\nu) \supseteq \{\mu + C, \nu + C' | (\mu,\nu) \in \arg\min_{\mu,\nu} \mathcal{L}(\mu,\nu), C \in \mathbb{R}, C' \in \mathbb{R}\}$$

$\square$

Remark that the proposed objective (51) is stable and the aforementioned proposition allows us to use self-normalized weighted importance sampling to extract policy. However, as we discussed in Section 3.5, using $\mu$ is main bottleneck for the overall optimization and optimizing $\tilde{\mathcal{L}}(\nu)$ shows better performance in practice.

# G Pseudocode of LobsDICE

## G.1 Tabular LobsDICE

For tabular MDPs, we first construct a maximum-likelihood estimation (MLE) MDP $\hat{M} = \langle S, A, \hat{T}, \hat{p}_0, \gamma \rangle$ using an offline dataset $D^I$. Then, tabular LobsDICE solves (16) on the MLE MDP $\hat{M}$. In tabular case, note that $\hat{d}^I(s, a)$, $\hat{d}^I(s, s')$, and $\hat{d}^E(s, s')$ are explicitly accessible by normalized counts of $(s, a)$ and $(s, s')$ in the datasets. In addition, $\hat{r}(s, s') = \frac{\hat{d}^E(s,s')}{\hat{d}^I(s,s')}$ is the log ratio between two empirical distributions. Also, $\nu \in \mathbb{R}^{|S|}$ is represented as a $|S|$-dimensional vector, and $\mu \in \mathbb{R}^{|S| \times |S|}$ is represented as a $|S|^2$-dimensional vector, and we solve the following convex minimization problem for $(\mu, \nu)$:

$$\min_{\mu, \nu} J(\mu, \nu) = (1 - \gamma) \sum_s \hat{p}_0(s) \nu(s) + \sum_{s,s'} \hat{d}^I(s, s') \Big[ \exp\big(\hat{r}(s, s') + \mu(s, s') - 1\big) \Big]$$
$$+ \alpha \sum_{s,a} \hat{d}^I(s, a) \Big[ \exp\Big(\tfrac{1}{\alpha} \sum_{s'} \hat{T}(s'|s, a)\big(-\mu(s, s') + \gamma \nu(s') - \nu(s)\big) - 1\Big) \Big] \quad (52)$$

This process is summarized in Algorithm 1.

---

**Algorithm 1** Tabular LobsDICE

---

**Input:** state-only demonstrations by experts $D^E = \{(s, s')_i\}_{i=1}^{N_E}$, state-action demonstrations by some imperfect agents $D^I = \{(s, a, s')_i\}_{i=1}^{N_I}$, a learning rate $\eta$.
 1: Construct an MLE MDP $\hat{M} = \langle S, A, \hat{T}, \hat{p}_0, \gamma \rangle$ via normalized visitation counts of $D^I$.
 2: Construct $\hat{d}^I(s, a)$, $\hat{d}^I(s, s')$, and $\hat{d}^E(s, s')$ via normalized visitation counts of $D^I$ and $D^E$.
 3: $\hat{r}(s, s') \leftarrow \log \frac{\hat{d}^E(s,s')}{\hat{d}^I(s,s')}$ for all $s, s'$.
 4: Randomly initialize $\nu$ and $\mu$.
 5: **while** $(\nu, \mu)$ is not converged **do**
 6: $\quad (\mu, \nu) = (\mu, \nu) - \eta \nabla_{\mu,\nu} J(\mu, \nu) \quad$ (Eq. (52))
 7: **end while**
 8: $w^*(s, a) = \exp\Big(\tfrac{1}{\alpha}\big(\sum_{s'} \hat{T}(s'|s, a)(-\mu(s, s') + \gamma \nu(s') - \nu(s))\big) - 1\Big)$ for all $s, a$. (Eq. (15))
 9: $\pi^*(a|s) \leftarrow \frac{\hat{d}^I(s,a)w^*(s,a)}{\sum_{a'} \hat{d}^I(s,a')w^*(s,a')}$ for all $s, a$.
**Output:** The imitation policy $\pi^*$.

---

## G.2 Practical LobsDICE (with function approximation)

To deal with continuous or large MDPs, we represent our optimization variable $\nu$ and discriminator $c$ as neural networks, parameterized by $\theta$ and $\phi$ respectively: $\nu_\theta : S \to \mathbb{R}$ is an MLP that takes a state as an input and outputs a scalar value, and $c_\phi : S \times S \to [0, 1]$ is defined similarly. For the policy $\pi_\psi$, we use a tanh-squashed Gaussian policy, where the parameters of a Gaussian distribution (i.e. mean and covariance) are output by the neural network.

The parameters of $c_\phi$ are trained by:

$$\max_\phi J_c(\phi) := \mathbb{E}_{\substack{\text{batch}(\bar{d}^E) \sim \bar{d}^E \\ \text{batch}(\bar{d}^I) \sim \bar{d}^I}} \Big[ \hat{\mathbb{E}}_{(s,s') \sim \text{batch}(\bar{d}^E)}[\log c_\phi(s, s')] + \hat{\mathbb{E}}_{(s,s') \sim \text{batch}(\bar{d}^I)}[\log(1 - c_\phi(s, s'))] \Big],$$
$$(53)$$

which is analogous to (13). The parameters of $\nu_\theta$ are trained by:

$$\min_\theta J_\nu(\theta) := \mathbb{E}_{\substack{\text{batch}(d^I) \sim d^I \\ \text{batch}(p_0) \sim \hat{p}_0}} \Big[ (1 - \gamma) \hat{\mathbb{E}}_{s \sim \text{batch}(p_0)}[\nu_\theta(s)] \qquad\qquad (54)$$
$$+ (1 + \alpha) \log \hat{\mathbb{E}}_{(s,a,s') \sim \text{batch}(d^I)}[\exp(\tfrac{1}{1+\alpha} \hat{A}_\theta(s, a, s'))] \Big],$$

where $\hat{A}_\theta(s, a, s') = r_\phi(s, s') + \gamma \nu_\theta(s') - \nu_\theta(s)$ and $r_\phi(s, s') = -\log\Big(\frac{1}{c_\phi(s,s')} - 1\Big)$, and this is analogous to (23). Due to the logarithm outside the expectation in the second term, the mini-batch

approximation would introduce additional bias in gradient estimation, but we found that using the biased gradient estimate worked well in practice with moderately large batch size (e.g. 256). Finally, the parameters of $\pi_\psi$ are trained by:

$$\max_\psi J_\pi(\psi) := \mathbb{E}_{\text{batch}(d^I) \sim d^I} \left[ \frac{\hat{\mathbf{E}}_{(s,a,s') \sim \text{batch}(d^I)} \left[ \exp\left(\frac{1}{1+\alpha} \hat{A}_\theta(s,a,s')\right) \log \pi_\psi(a|s) \right]}{\hat{\mathbf{E}}_{(s,a,s') \sim \text{batch}(d^I)} \left[ \exp\left(\frac{1}{1+\alpha} \hat{A}_\theta(s,a,s')\right) \right]} \right], \quad (55)$$

which is analogous to (24). Here, the self-normalized importance sampling may introduce an additional bias with mini-batch approximations, but it is well known that the self-normalized importance sampling provides a consistent estimator [31]. Also, it worked well in practice with moderately large batch size (e.g. 256) in our experiments. We optimize the parameters $(\phi, \theta, \psi)$ jointly in practice. We are not using a target network at all. The overall training process is summarized in Algorithm 2.

---

**Algorithm 2** LobsDICE (with function approximation)

---

**Input:** state-only demonstrations by experts $D^E = \{(s,s')_i\}_{i=1}^{N_E}$, state-action demonstrations by some imperfect agents $D^I = \{(s,a,s')_i\}_{i=1}^{N_I}$, a learning rate $\eta$.
1: Initialize parameter vectors $\phi, \theta, \psi$.
2: **for** each gradient step **do**
3:     Sample mini-batches from $D^E$ and $D^I$.
4:     Compute gradients and perform SGD update:
5:         $\phi \leftarrow \phi + \eta \nabla_\phi J_c(\phi)$         (Eq. (53))
6:         $\theta \leftarrow \theta - \eta \nabla_\theta J_\nu(\theta)$         (Eq. (54))
7:         $\psi \leftarrow \psi + \eta \nabla_\psi J_\pi(\psi)$         (Eq. (55))
8: **end for**
**Output:** The imitation policy $\pi_\psi$.

---

# H   Additional Experiments

## H.1   Subsampled expert demonstrations

Existing works on imitation learning [11] conduct experiments where the expert demonstrations are subsampled. The following figure presents the result when given the subsampled expert demonstrations. LobsDICE still significantly outperforms the baseline algorithms even with the subsampled expert demonstrations[3].

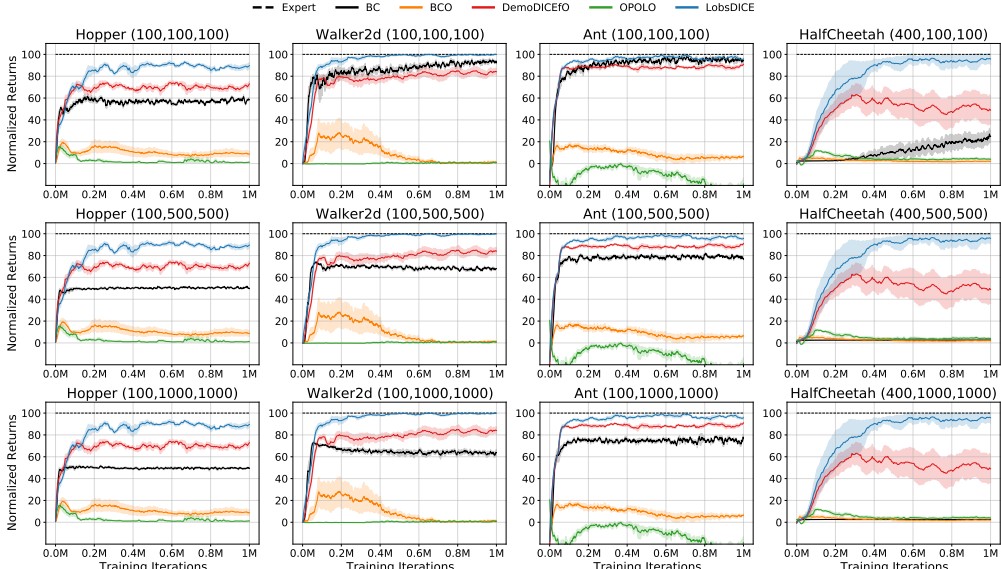

Figure 5: Performance of LobsDICE and baseline algorithms on various MuJoCo control tasks. We build state-only expert demonstrations using 50 subsampled trajectories from `expert-v2` (subsampling rate is 20).

## H.2   Error of inverse dynamics model in MuJoCo domains

In principle, IDM trained with arbitrary demonstrations should be able to predict the expert's missing actions accurately in deterministic MDPs. However, in Section 5.2, we observed that the empirical performance of BCO and DemoDICEfO tends to degrade as the number of imperfect demonstrations increases (see Figure 2). This is due to the use of function approximation for IDM: given that the expressive power of a function approximator is limited, the more data unrelated to the expert demonstrations (i.e. `medium-v2` and `random-v2`) is used for training, the more the prediction accuracy for the expert data would be adversely affected. Figure 6 shows that the mean squared error of IDM for expert demonstrations $D^E$ increases as the number of non-expert demonstrations increases.

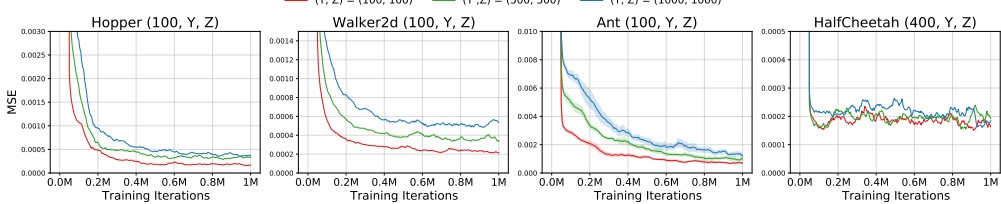

Figure 6: Mean squared error of IDM for expert demonstrations $D^E$. For each task $(Y, Z)$, we construct imperfect demonstrations using $100$ ($400$ for HalfCheetah), $Y$, and $Z$ trajectories from `expert-v2`, `medium-v2`, and `random-v2`, respectively.

---

[3]Recently, Li et al. reported that ValueDICE [17] (without target network) significantly underperforms DAC [16] (with target network) in the subsampled expert trajectories setting due to divergence issue and pointed to the absence of the target network in ValueDICE as a reason for performance degradation. However, we didn't observe any divergence issue due to the absence of the target network, and LobsDICE still performs well even when the trajectories are subsampled.

### H.3 Fewer expert demonstrations

In this section, we provide additional experiments with fewer expert demonstrations. Specifically, we tested LobsDICE when the number of state-only expert demonstrations is 5, 3, and 1, respectively. The Figure 7 shows that LobsDICE (#exp=3) and LobsDICE (#exp=1) still perform well.

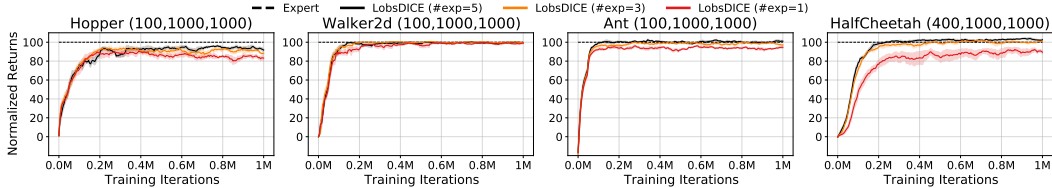

Figure 7: Performance of LobsDICE on various MuJoCo control tasks. We build state-only expert demonstrations using 1, 3, and 5 trajectories from expert-v2. For each task $(X, Y, Z)$ we construct imperfect demonstrations using $X$, $Y$, and $Z$ trajectories from expert-v2, medium-v2, and random-v2, respectively. We plot the mean and the standard errors (shaded area) of the normalized scores over five random seeds.

# I   Experimental Details

## I.1   Experimental details for random MDPs

**Random MDP generation**    For MDP $M = \langle S, A, T, R, \gamma, p_0 \rangle$, we first set $|S| = 20$, $|A| = 4$, $\gamma = 0.95$, $p_0(s) = 1$ for a fixed $s = s_0$. For each $(s, a)$, we samples four different state $\{s_1', s_2', s_3', s_4'\}$. Then, we set the transition probability $(T(s_1'|s, a), T(s_2'|s, a), T(s_3'|s, a), T(s_4'|s, a)) = (1 - \beta)X + \beta Y$, where $X \sim \text{Categorical}(0.25, 0.25, 0.25, 0.25)$ and $Y \sim \text{Dir}(1, 1, 1, 1)$. $\beta \in [0, 1]$ is the hyperparameter to control transition stochasticity: when $\beta = 0$, the transition probability becomes one-hot vector, i.e., deterministic MDP. In contrast, when $\beta = 1$, the transition of MDP becomes stochastic. Finally, the reward of 1 is only given to a state that minimizes the optimal value at initial state $s_0$; other states have zero rewards.

**Offline dataset generation**    For each random MDP $M$, we generate state-only expert demonstrations by executing a (softmax) expert policy and state-action imperfect demonstrations by a uniform random policy. We perform experiments for a varying number of expert demonstrations $N_E \in \{10, 100, 1000, 1000\}$ and imperfect demonstrations $N_I \in \{1, 3, 10, 30, 100, 300, 1000, 3000, 10000\}$.

**Hyperparameters**    We compare our tabular LobsDICE with BC, BCO, DemoDICEfO, and OPOLO. For KL-regularization hyperparameters of DemoDICEfO and LobsDICE, we use $\alpha = 0.1$.

## I.2   Experimental details for MuJoCo control tasks

| Hyperparameters | BC | BCO | DemoDICEfO | LobsDICE |
|---|---|---|---|---|
| $\gamma$ (discount factor) | 0.99 | 0.99 | 0.99 | 0.99 |
| $\alpha$ (regularization coefficient) | - | - | 0.1 | 0.1 |
| learning rate (actor) | $3 \times 10^{-4}$ | $3 \times 10^{-4}$ | $3 \times 10^{-4}$ | $3 \times 10^{-4}$ |
| network size (actor) | [256, 256] | [256, 256] | [256, 256] | [256, 256] |
| learning rate (critic) | - | - | $3 \times 10^{-4}$ | $3 \times 10^{-4}$ |
| network size (critic) | - | - | [256,256] | [256, 256] |
| learning rate (discriminator) | - | - | $3 \times 10^{-4}$ | $3 \times 10^{-4}$ |
| network size (discriminator) | - | - | [256,256] | [256, 256] |
| learning rate (inverse dynamics) | - | $3 \times 10^{-4}$ | $3 \times 10^{-4}$ | - |
| network size (inverse dynamics) | - | [256,256] | [256, 256] | - |
| gradient L2-norm coefficient (critic) | - | - | $1 \times 10^{-4}$ | $1 \times 10^{-4}$ |
| gradient penalty coefficient (discriminator) | - | - | 0.1 | 0.1 |
| batch size | 512 | 512 | 512 | 512 |
| # of expert trajectories | 5 | 5 | 5 | 5 |
| # of training iterations | 1,000,000 | 1,000,000 | 1,000,000 | 1,000,000 |

Table 1: Configurations of hyperparameters used in our experimental results.

For fair comparison, we use the same learning rate to train actors of BC, BCO, DemoDICEfO, and LobsDICE. We implement our network architectures for BC, BCO, DemoDICEfO, and LobsDICE based on the implementation of OptiDICE[4]. For OPOLO, we use its official implmentation[5]. We have tried hyperparameter tuning for OPOLO but never obtained a successful learning curve, showing numerical instability due to using out-of-distribution action values. Vanilla *online* off-policy algorithms commonly fail in the *offline* learning setting. Therefore, we report the results of official implementation without any modification to network architectures or hyperparameters. For stable discriminator learning, we use gradient penalty regularization on the $r(s, a)$ and $r(s, s')$ functions, which was proposed in [10] to enforce 1-Lipschitz constraint. To stabilize critic training, we add gradient L2-norm to the critic loss for the regularization. In addition, we use every state $s$ in $D^I$

---

[4] https://github.com/secury/optidice

[5] https://github.com/illidanlab/opolo-code

to define $D_0$, i.e., $D_0 = \{s|(s, a, s') \in D^I\}$, following ValueDICE [17]. Detailed hyperparameter configurations used for our main experiments are summarized in Table 1.

**Evaluation metric** For each environment, the normalized score is measured by $100 \times \frac{\text{score}-\text{random score}}{\text{expert score}-\text{random score}}$, where the `expert score` and `random score` are average returns of trajectories in `expert-v2` and `random-v2`, respectively.

## J   Generalization to $\gamma = 1$

In this section, we generalize LobsDICE for discounted MDPs ($\gamma < 1$) to undiscounted MDPs ($\gamma = 1$). Suppose that the stationary distributions $d$ and $\bar{d}$ satisfy the Bellman flow constraints (7) and marginalization constraints (8), respectively. When $\gamma = 1$, for any constant $c \geq 0$, $cd$ and $c\bar{d}$ also satisfy the corresponding constraints.

It means that the original constrained optimization problem (6-8) for the stationary distributions $d$ and $\bar{d}$ is an ill-posed problem. We tackle this ill-posedness by adding additional normalization constraint $\sum_{s,a} d(s, a) = 1$ to (6-8):

$$\max_{d,\bar{d} \geq 0} -D_{\text{KL}}(\bar{d}(s, s')\|\bar{d}^E(s, s')) - \alpha D_{\text{KL}}(d(s, a)\|d^I(s, a))$$

$$\text{s.t.} \sum_{a'} d(s', a') = (1-\gamma)p_0(s') + \gamma \sum_{s,a} d(s, a)T(s'|s, a) \quad \forall s',$$

$$\sum_{a} d(s, a)T(s'|s, a) = \bar{d}(s, s') \quad \forall s, s',$$

$$\sum_{s,a} d(s, a) = 1.$$

Using a derivation similar to that from (9) to (11), we obtain the following min-max optimization for $w, \bar{w}, \mu, \nu$, and $\lambda$:

$$\min_{\mu,\nu,\lambda} \max_{w,\bar{w} \geq 0} \mathcal{L}(w, \bar{w}, \mu, \nu, \lambda)$$

$$:= \mathcal{L}(w, \bar{w}, \mu, \nu) + \lambda(1 - \mathbb{E}_{(s,a) \sim d^I}[w(s, a)])$$

$$= (1-\gamma)\mathbb{E}_{s_0 \sim p_0}[\nu(s_0)] + \mathbb{E}_{(s,s') \sim \bar{d}^I}\left[\bar{w}(s, s')\big(r(s, s') + \mu(s, s') - \log \bar{w}(s, s')\big)\right]$$

$$+ \mathbb{E}_{(s,a) \sim d^I}\left[w(s, a)\big(\underbrace{e_{\mu,\nu}(s, a) - \lambda}_{:=e_{\mu,\nu,\lambda}(s,a)} - \alpha \log w(s, a)\big)\right] + \lambda, \tag{56}$$

where $\lambda \in \mathbb{R}$ is the Lagrange multiplier for the normalization constraint $\sum_{s,a} d(s, a) = 1$. Similar to Proposition 3.1, we can derive a closed-form solution to the inner maximization of (56):

$$w_{\mu,\nu,\lambda}(s, a) = \exp\left(\tfrac{1}{\alpha}e_{\mu,\nu,\lambda}(s, a) - 1\right) \text{ and } \bar{w}_\mu(s, s') = \exp(r(s, s') + \mu(s, s') - 1).$$

Then, we can reduce the nested min-max optimization of (56) to a single minimization by plugging the closed-form solution $(w_{\mu,\nu,\lambda}, \bar{w}_\mu)$ into $\mathcal{L}(w, \bar{w}, \mu, \nu, \lambda)$:

$$\min_{\mu,\nu,\lambda} \mathcal{L}(w_{\mu,\nu,\lambda}, \bar{w}_\mu, \mu, \nu, \lambda) = (1-\gamma)\mathbb{E}_{s \sim p_0}[\nu(s)]$$

$$+ \mathbb{E}_{(s,s') \sim \bar{d}^I}\left[\exp\left(r(s, s') + \mu(s, s') - 1\right)\right] + \alpha\mathbb{E}_{(s,a) \sim d^I}\left[\exp\left(\tfrac{1}{\alpha}e_{\mu,\nu,\lambda}(s, a) - 1\right)\right] + \lambda.$$

In practice, we estimate $\mathcal{L}(w_{\mu,\nu,\lambda}, \bar{w}_\mu, \mu, \nu, \lambda))$ using samples from distribution $d^I$. To derive a sample-based objective, we use analogous derivation in Section 3.5. Sample-based objective can be represented as

$$\min_{\mu,\nu,\lambda} \widehat{\mathcal{L}}(\mu, \nu, \lambda) = (1-\gamma)\hat{\mathbf{E}}_{s \in D_0}[\nu(s)] \tag{57}$$

$$+ \hat{\mathbf{E}}_{x \in D^I}\left[\exp\left(r(s, s') + \mu(s, s') - 1\right) + \alpha \exp\left(\tfrac{1}{\alpha}\hat{e}_{\mu,\nu,\lambda}(s, a, s') - 1\right)\right] + \lambda,$$

where $\hat{e}_{\mu,\nu,\lambda}(s, a, s') = -\mu(s, s') + \gamma\nu(s') - \nu(s) - \lambda$. Then, the closed-form solution to the minimization (57) with respect to $\mu$ is:

$$\mu_{\nu,\lambda}(s, s') = \frac{1}{1+\alpha}\big(-\alpha r(s, s') + \gamma\nu(s') - \nu(s) - \lambda\big) \tag{58}$$

Using the above solution, we obtain the following minimization problem:

$$\min_{\widehat{\nu},\widehat{\lambda}} \widehat{\mathcal{L}}(\widehat{\nu},\widehat{\lambda}) = (1-\gamma)\hat{\mathbf{E}}_{s\in D_0}[\nu(s)] + \widehat{\lambda} + (1+\alpha)\hat{\mathbf{E}}_{x\in D^I}\left[\exp\left(\frac{1}{1+\alpha}\widehat{A}_{\widehat{\nu},\widehat{\lambda}}(s,a,s')-1\right)\right], \quad (59)$$

where $\widehat{A}_{\nu,\lambda}(s,a,s') := r(s,s') + \gamma\nu(s') - \nu(s) - \lambda$. Finally, similar to Proposition 3.4, we can obtain the following objective with the same minimum as (59):

$$\min_{\nu,\lambda} \widetilde{\mathcal{L}}(\widetilde{\nu},\widetilde{\lambda}) = (1-\gamma)\hat{\mathbf{E}}_{s\in D_0}[\widetilde{\nu}(s)] + \widetilde{\lambda} + (1+\alpha)\log\hat{\mathbf{E}}_{x\in D^I}\left[\exp\left(\frac{1}{1+\alpha}\widehat{A}_{\widetilde{\nu},\widetilde{\lambda}}(s,a,s')\right)\right]$$

$$= (1-\gamma)\hat{\mathbf{E}}_{s\in D_0}[\widetilde{\nu}(s)] + (1+\alpha)\log\hat{\mathbf{E}}_{x\in D^I}\left[\exp\left(\frac{1}{1+\alpha}\widehat{A}_{\widetilde{\nu}}(s,a,s')\right)\right] = \min_{\nu} \widetilde{\mathcal{L}}(\widetilde{\nu}).$$

Interestingly, the resulting objective is the same as the original objective $\widetilde{\mathcal{L}}(\widetilde{\nu})$ of (23). Based on this theoretical result, we compare LobsDICE for $\gamma = 1$, denoted by LobsDICE ($\gamma = 1$), with LobsDICE for $\gamma = 0.99$, denoted by LobsDICE ($\gamma = 0.99$). In Figure 8, LobsDICE ($\gamma = 1$) shows good performance, but LobsDICE ($\gamma = 0.99$) is more stable and achieves better performance in practice.

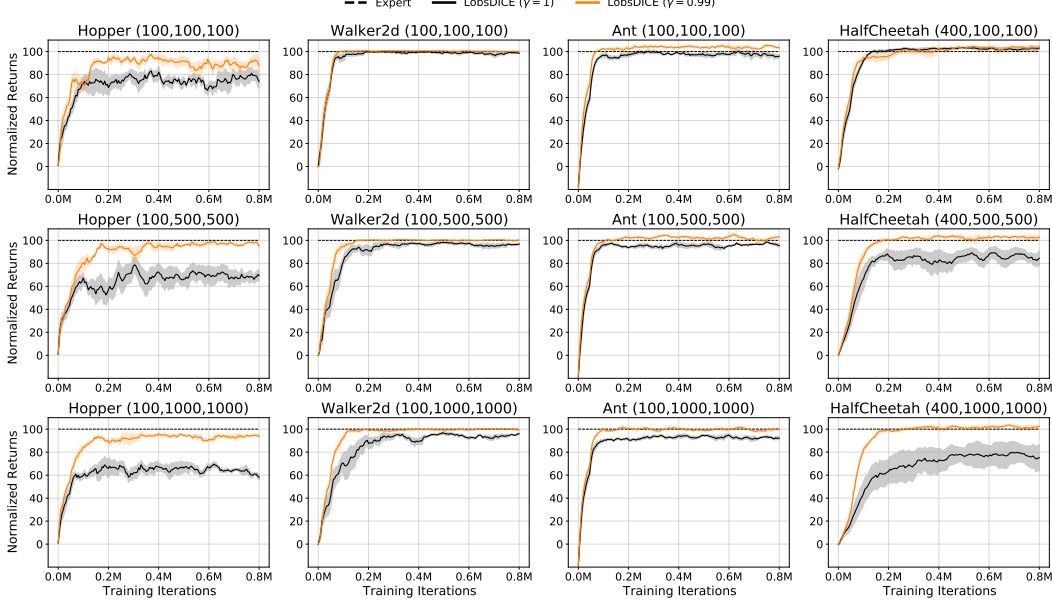

Figure 8: Performance of LobsDICE ($\gamma = 0.99$) and LobsDICE ($\gamma = 1$) on various MuJoCo control tasks. We build state-only expert demonstrations using 5 trajectories from `expert-v2`. For each task $(X, Y, Z)$ we construct imperfect demonstrations using $X$, $Y$, and $Z$ trajectories from `expert-v2`, `medium-v2`, and `random-v2`, respectively. We plot the mean and the standard errors (shaded area) of the normalized scores over five random seeds.

## K    Computation Resources

We used 10 servers equipped with the following specification:

- CPU: Intel(R) Core(TM) i7-9700K CPU @ 3.60GHz.
- Memory: 32 GB.
- GPU: TITAN V.

## L    Limitation

Although we demonstrated that LobsDICE successfully recovered the expert's behavior in the experiments, it requires the assumption that support of $D^I$ covers $D^E$. Relaxing this assumption remains as a future work.

# M License

D4RL [6] is licensed under the Apache 2.0

# N Social Impact

This work contributes to an algorithmic foundation for imitation learning from observation in an offline setting. Given that state-only expert demonstrations are much easier to be collected than the action-labeled expert demonstrations and that the imperfect demonstrations of arbitrary optimality is also easy to be collected, our method has a potential to be widely adopted in many real-world applications. On the other hand, this work could adversely affect employment by contributing to the automation of tasks having done by human experts (e.g. factory automation, autonomous driving).