# OpenReview forum: "LobsDICE: Offline Learning from Observation via Stationary Distribution Correction Estimation"
_NeurIPS.cc/2022/Conference — NeurIPS 2022 Accept_

### Official Review · Reviewer_zkgU · 2022-07-12

**Rating:** 7
**Confidence:** 3
**Soundness:** 4 excellent
**Presentation:** 3 good
**Contribution:** 3 good

**Summary:**

The authors propose an algorithm for offline learning from _observations_, i.e. for learning from observed expert trajectories where only the states, not the actions can be observed. Additionally, similar to some prior work, the authors assume access to a large set of data from suboptimal policies, but containing actions.

Overall, the paper follows the DemoDICE paper very closely, both in structure of the paper as well as in terms of the algorithm, but with the crucial difference than DemoDICE assumes access to expert actions.

Interestingly, the optimization target for both algorithms ends up looking exactly alike:

Equation (23) from LobsDice:

$$
\widetilde{\mathcal{L}}(\widetilde{\nu})=(1-\gamma) \hat{\mathbf{E}}\_{s\_{0} \in D_{0}}\left[\widetilde{\nu}\left(s_{0}\right)\right]+(1+\alpha) \log \hat{\mathbf{E}}\_{x \in D^{I}}\left[\exp \left(\frac{1}{1+\alpha} \widehat{A}_{\widetilde{\nu}}\left(s, a, s^{\prime}\right)\right)\right]
$$

Equation (17) from DemoDice:

$$
\widetilde{L}(\nu ; r):=(1-\gamma) \mathbb{E}\_{s \sim p\_{0}}[\nu(s)]+(1+\alpha) \log \mathbb{E}\_{(s, a) \sim d^{U}}\left[\exp \left(\frac{A\_{\nu}(s, a)}{1+\alpha}\right)\right]
$$

with, as far as I can tell, the only difference being whether the discriminator is learned on tuples $(s,s')$ or $(s,a)$.

However, as far as I can tell, it is not obvious a-priori that the solutions will end up looking the same, hence making the derivations in the paper necessary.

Performance is evaluated against several baselines (BC, BCO, DemoDICE with inverse dynamics model, OPOLO) on both a tabular environment and several MuJoCo environments (Hopper, Walker2d, Ant and HalfCheetah), showing good performance.

**Questions:**

Please see my suggestions above regarding small changes to the presentation of the paper.
I don't have any question and I think this is a solid paper which should be accepted.
I'm not rating it stronger than "accept", due to the similarity (in approach and solution) to DemoDICE.

**Limitations:**

N/A.

**Strengths And Weaknesses:**

## Strenghts

* I found the writing accessible and good to follow, as well as the mathematical derivation well explained. I do think the paper might be a bit harder to follow if one is not familiar with similar algorithms (e.g. the *DICE family), so a short recap of their their core ideas in the background section would be helpful. Similarly, a recap of DemoDICE would be helpful and make the close connection clearer.
* The topic of offline IL is relevant, as is the extension to LfO
* I liked the experimental section and found both the figures and discussion easy to follow as well as insightful.

## Weaknesses

* I think the presentation of the derivation of the algorithm could be slightly improved. While each step was easy to follow, it took me a while (including going back and forth between LobsDICE and other *DICE papers) to get a decent high level motivation for the individual steps. It certainly does not require any major changes, but maybe a bit of additional signposting could help.
* I think the relationship to DemoDICE should be made clearer. The authors already mention that DemoDICE solves a related problem, but I would have found it helpful if also the similarities and differences in the solution would be discussed.
* One could argue that the "delta" in terms of research compared to DemoDICE is limited. However, I believe there is sufficient difference to warrant publication.

---

> ### Author Response · Authors · 2022-08-02
> **Response to Reviewer zkgU**
>
> We appreciate your constructive feedback.
>
> **[High level motivation of the derivation of LobsDICE]**
>
> Our original formulation (6-8) naturally describes the state-transition stationary distribution matching problem in the space of stationary distribution. However, solving this constrained optimization problem in its form is challenging, except for small finite MDPs. To make optimization tractable using stochastic gradient descent in the offline setting, the objective function should be expressed in terms of expectation with respect to the offline data (i.e. either $d^I$ or $d^E$). To this end, we first consider the Lagrangian (9), which converts the constrained optimization into the unconstrained one. Then, we rearrange the terms so that every term is expressed in terms of expectation w.r.t. $d$ or $\bar d$ (10). Finally, we adopt change-of-variable (e.g. $w(s,a)=d(s,a)/d^I(s,a)$) so that every expectation is expressed in terms of the dataset distribution (11). We will add more explanations in the camera-ready version of the paper using an additional page.
>
> **[Similarities and differences between DemoDICE and LobsDICE]**
>
> As a DICE-family algorithm, LobsDICE has similarities with DemoDICE in algorithm derivation. The final sample-based objective function of LobsDICE (23) is identical to that of DemoDICE, except that the discriminator is learned on $(s,s’)$ instead of $(s,a)$.
> However, obtaining this result by naively extending DemoDICE is *not* straightforward. Note that DemoDICE is an algorithm that optimizes the state-action stationary distribution $d(s,a)$, where its application to $(s,s')$-distribution matching problem yields: $D_\mathrm{KL}(\bar d(s,s') || \bar d^E(s,s')) = \mathbb{E}\_{(s,a) \sim d, s' \sim T(s,a)} \Big[ \log \frac{ \sum\_{a} d(s,a) T(s'|s,a) }{d^E(s,s')} \Big]$. This computation is intractable in the offline setting. Please refer to Appendix D for more discussions on the challenge of extending DemoDICE to offline LfO.

---

> > ### Comment · Reviewer_zkgU · 2022-08-06
> > **Thank you for your reply.**
> >
> > Thank you for your reply. I agree that the extension to DemoDICE is not straightforward, hence my vote for acceptance.

---

### Official Review · Reviewer_s7rN · 2022-07-12

**Rating:** 8
**Confidence:** 3
**Soundness:** 4 excellent
**Presentation:** 3 good
**Contribution:** 3 good

**Summary:**

This paper algorithmically extends recent advances to the DICE literature in the offline RL setting to the IFO setting. They present a novel extension to match state-transition distributions as well as a more stable minimization objective to achieve SOTA performance for Offline IFO on the Mujoco benchmark task.

**Questions:**

This is not a critique, but from the experiments it looked like you used 5 expert demonstrations. I'm curious about how low could you go? BC is able to achieve expert performance with as low as 1 trajectory in these environments (I understand BC also gets actions so has a definite advantage), but it would be really cool to see how LobsDICE does with less and less expert demonstrations.

**Limitations:**

Yes the authors have sufficiently addressed limitations and societal impacts in their paper.

**Strengths And Weaknesses:**

### Strengths
1. Technical contribution: Extends much of the recent advancements in the DICE literature to the IFO setting to do (s, s') distribution matching.
2. Definitely excited to see the stabilization of the DICE objective to be a single minimization than a min-max. On a side note, this makes me wonder if this sheds some light into scaling up to higher dimensional benchmarks?
3. Strong experimentation showing the effective state-transition minimization with state-of-the-art performance on the Mujoco benchmarks. I know D4RL does not provide datasets for Humanoid but would be interesting to fully finish up the benchmark suite for IFO.

### Weaknesses
1. One small nitpick would be compare against an Offline IL method such as MILO [1] slightly modified for IFO (train discriminator on (s, s') rather than (s, a)). This would be a comparison against a principled offline method for adversarial IL. This would provide more proof of the stability contribution of this work of having just a minimization objective rather than a min-max objective - a benefit the authors mention throughout the text.


[1] Mitigating Covariate Shift in Imitation Learning via Offline Data Without Great Coverage, 2021

---

> ### Author Response · Authors · 2022-08-02
> **Response to Reviewer s7rN**
>
> We appreciate your constructive feedback.
>
> **[Comparison with MILO]**
>
>  MILO cannot be easily extended to the offline LfO setting due to its using behavior cloning on the expert dataset as a regularization, i.e. we don't have access to expert's actions in the LfO setting. Without the BC loss, we have never observed the successful results of MILO in our experiments using D4RL datasets.
>
> **[Using fewer expert demonstrations]**
>
> We provide additional experiments with fewer expert demonstrations in Appendix H.3. We tested LobsDICE with fewer expert demonstrations (3 and 1), and their performance drops are marginal (Figure 7).

---

> > ### Comment · Reviewer_s7rN · 2022-08-08
> > **Thanks for the reply!**
> >
> > Thanks for the response. I will keep my score unchanged.

---

### Official Review · Reviewer_iQ5k · 2022-07-12

**Rating:** 6
**Confidence:** 3
**Soundness:** 3 good
**Presentation:** 3 good
**Contribution:** 2 fair

**Summary:**

This paper propose an offline learning from observation (LfO) algorithms, which imitates the expert policy by stationary distribution matching without interacting with the environment. The authors presented how to transfer the original complex minimax optimization problem to a simple convex minimization problem in detail, which avoids the instability issue in the original DICE based algorithms.
Empirical experiments show the effectiveness of the proposed algorithms and shows competitive performance compared with baseline algorithms.

**Questions:**

I have a long existing fundamental question for this line of works, which try to use DICE techniques to perform offline imitation learnings, ever since ValueDICE, and also for this work.

 If we consider Equation (1) or final Equation (5) in the paper for offline imitation learning, what is the distribution of $\bar{d}^{E}(s, s^\prime)$, $d^{I}(s,a)$ or $d^{E}(s,a)$? Are they the **discounted** stationary distributions (the distribution would involve initial state distribution and a discounted factor $\gamma < 1$) or just the **average** visitation stationary distribution ($\gamma =1$ or the distribution you just uniformly sample from the empirical dataset).

This question is important, because when we consider to use the **discounted** stationary distribution $\bar{d}(s, s^\prime)$ or $d(s, a)$ of the current learning policy to match the target distributions ($d^{I}(s, a)$ or $\bar{d}^{E}(s, s^\prime)$), the target distributions should also be the **discounted** ones. As a result, $d^{I}(s, a)$ or $\bar{d}^{E}(s, s^\prime)$ must be the **discounted distribution**, which are not the **same as the distributions we uniformly sample from the empirical dataset**.

From what the authors describe in the paper, I think the authors just use the empirical data distribution as $d^{I}(s, a)$ or $\bar{d}^{E}(s, s^\prime)$, which are actually not right because the uniformly sampling procedure would give us the average visitation stationary distribution ($\gamma = 1$).

There are two ways to fix the flaw:
1. Consider to define $\bar{d}(s, s^\prime)$ or $d(s, a)$  as the average visitation stationary distribution ($\gamma = 1$), thus Equation (7) is different, and the whole derivation of this paper need to change.

2. Consider to change the sampling procedure such that $d^{I}(s, a)$ or $\bar{d}^{E}(s, s^\prime)$ are the discounted stationary distribution, which involves the initial distribution $p_{0}(s)$. One possible simple solution is to sample transition data according to $\gamma^t$, where $t$ is the time step.

-----
Empirically I think there might not be significant impact of above concern, and please correct me if I misunderstand anything in the paper, and I will change my score.










**Ethics Review Area:**

["I don’t know"]

**Limitations:**

The authors adequately addressed the limitations and there are no negative societal impact.

**Strengths And Weaknesses:**

- **Originality**.
The proposed algorithm is based on previous DICE techniques including DualDICE, OptiDICE for offline policy evaluation or improvement, and the authors extended the idea to the offline learning from observations settings. As far as I know, the techniques are not new but there are very few prior works that consider extending the idea to the offline LfO setting.

-  **Quality**.
The paper is technically sound and the empirical experiments are conducted thoroughly. One concern I have is that, as the authors mentioned, there are some concurrent works (SMODICE) related to this topic. In the appendix the authors discussed potential drawbacks of the concurrent work, while there are no empirical experiments to validate and support the claim, which I think the authors should make clear comparison and discussion in both the experiments and the related work sections (I checked with their paper, it seems that they already release the code?).

- **Clarity**.
Overall the paper is well presented and it is easy to follow. The derivation presented in the main content are easy to follow if the readers know the basic DICE techniques in the previous work.

- **Significance**.
The presented techniques are a little bit incremental, but I think the proposed algorithm is a good addition to the offline LfO settings.

---

> ### Author Response · Authors · 2022-08-02
> **Response to Reviewer iQ5k**
>
> We appreciate your constructive feedback.
>
> **[Comparison with SMODICE]**
>
> Although SMODICE, a concurrent work with ours, also aims to solve offline LfO via a DICE-based approach, it relies on a (potentially loose) upper bound. Furthermore, unlike OPOLO's upper bound gap of (4), the SMODICE's upper bound gap of $D\_\mathrm{KL}(\pi(a|s) || \pi^I(a|s))$ does not vanish even for deterministic MDPs, thus SMODICE is essentially suboptimal unless $D^I$ is purely collected by experts. To see this, we conducted an additional experiment for tabular SMODICE, presented in Figure 4. SMODICE significantly underperforms LobsDICE and baseline algorithms, as expected. Please refer to Remark B.1. in Appendix B for more discussions.
>
> **[Question on discounted stationary distribution]**
>
> As the reviewer suggested in (2), sampling from $\bar d^E(s,s'), d^I(s,a), d^E(s,a)$ should be done with a probability proportional to $\gamma^t$ in principle. However, this sampling strategy shows poor empirical performance since it would rarely sample the ones corresponding to later timesteps. Thus, in our experiments, we are using 'uniform' sampling (rather than 'discounted' sampling) for $\bar d^E(s,s'), d^I(s,a), d^E(s,a)$, which performs better in practice. For the sake of empirical performance, we have compromised some gaps between theory and practical implementation.
> Nevertheless, to close this gap, we have additionally derived an algorithm for $\gamma=1$ in Appendix J. To deal with $\gamma=1$, an additional normalization constraint $\sum\_{s,a} d(s,a) = 1$ should be considered in (6-8). Interestingly, our sample-based objective of (23) does not change even with the introduction of the normalization constraint in the derivation.
> We conducted additional experiments for LobsDICE with $\gamma=1$, presented in Figure 8 in Appendix J.

---

> > ### Comment · Reviewer_iQ5k · 2022-08-09
> > **Thank you for the Response**
> >
> > I really appreciate the authors for additional comparison and the extension to the $\gamma=1$ case, and I will raise my score.
> >
> > For the poor empirical performance of using the new sampling strategy, I think it is expected that the empirical performance may not be better than the original uniform sampling strategy in the discounted case because most of the hyperparameters are tunned based on the old one.
> >
> > However, I do think current offline imitation learning algorithms that are DICE based are problematic because the issue is totally ignored by the community and few works are trying to fix it. I think the authors should discuss this in the next version or in the future to draw people's attention, so the theoretical derived algorithms match the practical implementations.

---

### Official Review · Reviewer_cB6F · 2022-07-20

**Rating:** 4
**Confidence:** 4
**Soundness:** 3 good
**Presentation:** 3 good
**Contribution:** 2 fair

**Summary:**

This paper deals with offline LfO with imperfect demonstrations. The main contribution of this paper includes solving a distribution matching problem without  requiring an inverse dynamics model. The reduction of the problem is simple. But the algorithm and the theory heavily borrow from DemoDICE[15]. and the contribution of this paper is incremental.

**Questions:**

Figure 1 shows that the performance gap between LobsDICE and other baselines becomes larger as the level of stochasticity increases. And there is no gap when the environment is nearly deterministic.

However, the Mujoco environments in Figure 2 are deterministic within my knowledge. So how to explain the performance of LobsDICE in figure 2?

**Strengths And Weaknesses:**

- Strengths
  - The paper reduces the regular minimax problem in matching state distribution to a single convex minimization problem to make the proposed algorithms more stable than prior work.
  - There are sufficient baselines in the experiments to support the performance of LobsDICE.

- Weaknesses
  - The algorithm and the theory heavily borrow from DemoDICE[15].

---

> ### Author Response · Authors · 2022-08-02
> **Response to Reviewer cB6F**
>
> Thank you for your thoughtful comments.
>
> **[Novelty of LobsDICE, compared with DemoDICE]**
>
> As a DICE-family algorithm, LobsDICE has similarities with DemoDICE in algorithm derivation, but extending DemoDICE to the LfO setting is *not* straightforward. Note that DemoDICE is an algorithm that optimizes the state-action stationary distribution $d(s,a)$, where its application to $(s,s')$-distribution matching problem yields: $D_\mathrm{KL}(\bar d(s,s') || \bar d^E(s,s')) = \mathbb{E}\_{(s,a) \sim d, s' \sim T(s,a)} \Big[ \log \frac{ \sum\_{a} d(s,a) T(s'|s,a) }{d^E(s,s')} \Big]$. This computation is intractable in the offline setting. Please refer to Appendix D for more discussions.
>
> **[Experiments on MuJoCo]**
>
> Please refer to lines 298-319. OPOLO and IQ-Learn suffer from severe numerical instability due to the evaluation of OOD action values. The performance of BCO and DemoDICEfO is affected by the quality of the learned IDM  Although IDM should be able to successfully recover the expert action in deterministic MDPs in principle, we empirically observed that IDM's prediction performance degrades as the number of suboptimal trajectories in $D^I$ increases (Appendix H.2). Consequently, BCO and DemoDICEfO suffer from performance degradation.

---

### Author Response · Authors · 2022-08-02
**General Response**

We thank all the reviewers for their constructive comments and feedback. We summarize the important improvements and modifications in the revision as follows:

- We added additional experiments for tabular SMODICE [1] in Remark B.1. in Appendix B.
- We introduced generalization to $\gamma=1$ in Appendix J.
- We conducted additional experiments with fewer expert demonstrations in Appendix H.3.

[1] Ma, Yecheng, et al. "Versatile Offline Imitation from Observations and Examples via Regularized State-Occupancy Matching." ICML, 2022.

---

### Meta-Review · Area_Chair_wMsQ · 2022-08-26

**Recommendation:** Accept
**Confidence:** Certain

**Metareview:**

Reviewers all agree that the paper has solid contributions on both theory and experiments.

**Award:**

No

---

### Decision · Program_Chairs · 2022-09-14

Accept